# Reducing competition between *msd* and genomic DNA improves retron editing efficiency

Yuyang Ni [1,2,3], Yifei Wang [1,3], Xinyu Shi[1], Fan Yu[1], Qingmin Ruan[1], Na Tian[1], Jin He [1] & Xun Wang [1✉]

## Abstract

**Retrons, found in bacteria and used for defense against phages, generate a unique molecule known as multicopy single-stranded DNA (msDNA). This msDNA mimics Okazaki fragments during DNA replication, making it a promising tool for targeted gene editing in prokaryotes. However, existing retron systems often exhibit suboptimal editing efficiency. Here, we identify the *msd* gene in *Escherichia coli*, which encodes the noncoding RNA template for msDNA synthesis and carries the homologous sequence of the target gene to be edited, as a critical bottleneck. Sequence homology causes the msDNA to bind to the *msd* gene, thereby reducing its efficiency in editing the target gene. To address this issue, we engineer a retron system that tailors msDNA to the leading strand of the plasmid containing the *msd* gene. This strategy minimizes *msd* gene editing and reduces competition with target genes, significantly increasing msDNA availability. Our optimized system achieves very high retron editing efficiency, enhancing performance and expanding the potential for in vivo techniques that rely on homologous DNA synthesis.**

**Keywords** Gene Editing; Retron; DNA Replication; *Escherichia coli*
**Subject Categories** Biotechnology & Synthetic Biology; DNA Replication, Recombination & Repair; Methods & Resources

## Introduction

Single-stranded DNA (ssDNA) has emerged as a promising tool for gene editing (van der Oost and Patinios, 2023; Wannier et al, 2021; Yu et al, 2000). They contain flanking sequences (homology arms) and a central region. The flanking sequences are homologous to the target region in the genome and are typically 30-100 base pairs long, while the central region contains the desired genetic modifications, such as nucleotide changes, insertions, or deletions. These modifications in ssDNA enable desired genome editing in a wide variety of bacteria (Murphy, 2016; Sharan et al, 2009; Zhang et al, 2000). The "annealing-integration" model is widely recognized as a ssDNA-mediated gene editing mechanism. In this model, ssDNA-binding proteins facilitate the alignment of the ssDNA donor with its complementary strand near the replication fork, creating a heteroduplex that includes a mutated strand. As this heteroduplex is replicated, both mutated and wild-type progeny are produced (Ellis et al, 2001; Huen et al, 2006; Thomason et al, 2016). Reports from several labs suggest that ssDNA editing displays strand preference, whereby ssDNA with the same sequence in the nascent lagging strand (Okazaki fragments) produces a higher editing frequency than the leading strand (Costantino and Court, 2003; Ellis et al, 2001; Li et al, 2003). This strand bias can be explained by the presence of more single-stranded regions of DNA during lagging-strand synthesis than during leading-strand synthesis, to which ssDNAs have access (Ellis et al, 2001; Huen et al, 2006).

Retrons are unique reverse transcription systems found in various microbial genomes (Gonzalez-Delgado et al, 2021; Inouye and Inouye, 1993; Lampson et al, 2005; Mestre et al, 2020). One example is the Retron-Eco1 (Ec86) system discovered in *Escherichia coli* strain BL21. The retron operon consists of three genes: *msr/msd* (for the noncoding RNA, ncRNA), *ret* (for the reverse transcriptase, RTase) and *ndt* (for the ribosyltransferase effector protein) (Lim and Maas, 1989; Mestre et al, 2020; Millman et al, 2020). The ncRNA itself has two regions: msr and msd. RTase recognizes the structured ncRNA and uses the 2'-OH of the conserved guanosine residue in msr as a primer. It then reverse-transcribes the msd sequence, producing a hybrid RNA-ssDNA molecule called msDNA (Lampson et al, 2005; Lim and Maas, 1989). During this process, endogenous RNase H1 specifically digests the msd sequence (Lampson et al, 1989; Palka et al, 2022).

The discovery that the msDNA from the retron system can be used as-synthesized ssDNA in vivo for gene editing has attracted widespread attention. The *msd* in the retron operon serves as a template for msDNA production and is also the region where exogenous sequences are inserted during gene editing (Farzadfard et al, 2021; Farzadfard and Lu, 2014; Lear and Shipman, 2023; Lopez et al, 2022; Kaur and Pati, 2024; Simon et al, 2019; Simon et al, 2018; Tang and Sternberg, 2023). During the editing process, msDNA is integrated into the lagging strand of the DNA replicon in the form of Okazaki fragments by the same mechanism as ssDNA (Ellis et al, 2001; Gallagher et al, 2014; Huen et al, 2006).

[1]National Key Laboratory of Agricultural Microbiology, College of Life Science and Technology, Huazhong Agricultural University, Wuhan 430070, P. R. China. [2]College of Life Sciences, Shangrao Normal University, Shangrao 334001, P. R. China. [3]These authors contributed equally: Yuyang Ni, Yifei Wang. ✉E-mail: wangxun@mail.hzau.edu.cn

Researchers have been actively seeking to improve the efficiency of retron-mediated gene editing. The gene editing efficiency of the first-generation system in *E. coli* DH5α strain is less than $10^{-4}$ after ~30 generations of induction (24 h) (Farzadfard and Lu, 2014). The second generation involved introducing mutations in the $3'$-$5'$ exonuclease ExoX and the mismatch repair protein MutS into *E. coli* Top 10 strains. Following ~20 generations (16 h) of induction, the retron system's editing efficiency improved 100-fold to $6 \times 10^{-2}$ (Simon et al, 2018). In the third generation, Schubert et al (2021) significantly improved editing efficiency to nearly 100% after ~20 generations of induction and batch growth. This was achieved in the EcNR1 strain by knocking out the genes *mutS*, *recJ* (encoding a 5'-3' exonuclease), and *sbcB* (encoding a 3'-5' exonuclease) and replacing the single-stranded annealing protein Beta recombinase with CspRecT, another single-stranded annealing protein. Other studies have demonstrated that knocking out *recJ* and *sbcB* in DH5α strains, combined with the overexpression of Beta recombinase, can result in editing efficiencies nearing 100% in ~20 generations (about 16 h) (Liu et al, 2023) or 60 generations (around two days) (Farzadfard et al, 2021). Additionally, a recent study screened for new retrons and identified several that outperform Eco1 in efficiency (Khan et al, 2024). This research highlights significant diversity in msDNA production and editing rates among retrons. These editors exceed those used in previous studies, achieving precise editing rates of up to 40% in human cells.

Researchers are also working to modify ncRNAs to improve editing efficiency. They have explored how donor DNA homology arm length, msDNA secondary structure, and the strength of retron operon expression influence editing efficiency. Their findings show that optimizing these factors and increasing msDNA expression can significantly enhance editing efficiency (Bhattarai-Kline et al, 2022; Liu et al, 2023; Lopez et al, 2022; Schubert et al, 2021). Interestingly, they revealed a surprising effect of plasmid copy number on editing efficiency. When msDNA was expressed with different plasmids-p15A (5–10 copies), pBR322 (20–40 copies), pUC (400–600 copies), the editing efficiency varied after 20 generations (16 h) of retron system induction. The highest efficiency (29%) was observed with the lowest copy number plasmid (p15A), while it dropped to 13% with pBR322 and further down to 7% with the high copy number pUC plasmid (Liu et al, 2023). This suggests that unknown factors, beyond just the intensity of msDNA expression, significantly impact editing efficiency.

# Results and discussion

## Construction and evaluation of a retron editing system

We constructed a retron editing system using a plasmid derived from the p15A origin of replication. The retron operon, consisting of *msr*, *msd*, and *ret* genes, was placed under a strong constitutive promoter (J23119) and co-expressed with the CspRecT protein coding sequence to enhance editing efficiency (Schubert et al, 2021) (Fig. EV1A). The msDNA programmable region was modified with 92 bp homologous sequences targeting either strand of *lacZ*, denoted as PSlacZ-anti and PSlacZ in Fig. EV1A,B, respectively (where PS stands for partial sequence). Appendix Figs. S1, S2 provide additional details on the specific locations of these modifications. These 92 bp homologous sequences carry mutations that, upon successful editing, alter the codons for tryptophan and glutamate at positions 17 and 18 of the LacZ protein, converting TGG and GAA to termination codons (TGA and TAA). Colonies

with successfully edited *lacZ* gene will appear white on LB plates containing IPTG and X-Gal.

We evaluated the retron editing system in *E. coli* strains with varying mismatch repair deficiencies (MG1655, Δ*mutS*, Δ*mutSrecJ*, Δ*mutSrecJsbcB*), each harboring either the p15AlacZ or p15AlacZ-anti plasmid. The editing efficiency was minimal in all strains expressing msDNA-PSlacZ-anti (matching the leading strand). Conversely, msDNA-PSlacZ (matching the lagging strand) demonstrated increasing editing efficiency, reaching nearly 100% in the Δ*mutSrecJsbcB*-p15AlacZ strain (Fig. EV1B). Sanger sequencing confirmed successful editing in the Δ*mutSrecJsbcB*-p15AlacZ strain (Fig. EV1C). Our results demonstrate that msDNA efficiently edits the target gene only when the homologous sequence matches the lagging strand of genomic DNA replication, not the leading strand.

To investigate the effect of plasmid copy number on editing efficiency, we inserted the retron operon encoding msDNA-PSlacZ into two newly constructed plasmids: pSC101 (low copy) and pUC (high copy), naming them pSClacZ and pUClacZ, respectively (Fig. 1A; Appendix Figs. S3–S4). The previously constructed p15AlacZ plasmid (medium copy) was also included for comparison. The Δ*mutSrecJsbcB* strain was transformed with each of the above plasmids, and the editing efficiencies were recorded every 2 h from 2 h to 24 h. As shown in Fig. 1B, each strain exhibits a different editing efficiency. The editing efficiency of Δ*mutSrecJsbcB*-p15AlacZ approached 100% after 12 h of incubation. The editing efficiency of Δ*mutSrecJsbcB*-pSClacZ continued to increase with the extension of incubation time, reaching more than 90% after 24 h. However, the strain with a high-copy plasmid (Δ*mutSrecJsbcB*-pUClacZ) had the lowest editing efficiency, with the highest efficiency not exceeding 20% after 24 h of incubation (Fig. 1B). We then quantified plasmid and msDNA abundance at a stable editing time point. The results showed that plasmid copy numbers increased from pSClacZ (6) to p15AlacZ (50) to pUClacZ (420). Correspondingly, the msDNA copy numbers in these strains were 3, 155, and 340, respectively (Fig. 1C). These findings suggest that although msDNA production increases with plasmid copy number, it does not directly correlate with editing efficiency.

## The presence of plasmid *msd* reduced genomic DNA editing efficiency

To understand the reduced editing efficiency observed in the Δ*mutSrecJsbcB*-pUClacZ strain, we considered the mechanism of the retron editing system. We noted that msDNA is synthesized using the retron operon as a template. This situation could create potential competition within the cell, as the synthesized msDNA could target two locations: (1) the *msd* sequence located in the retron operon on the plasmid, which is the template sequence used for msDNA synthesis and (2) the genomic DNA, which serves as the target for genome editing. Therefore, during the editing process, the plasmid-based template *msd* might compete with the genomic DNA for binding to the msDNA (Fig. 2A). If this competition exists, increasing the plasmid copy number would lead to more *msd* competing with the genomic target. This would reduce the amount of msDNA available for editing the genomic DNA, ultimately reducing the editing efficiency.

To demonstrate that additional homologous sequences on the plasmid can reduce editing efficiency, we inserted another copy of the 92 bp *lacZ* sequence into plasmid p15AlacZ, creating p15AlacZ-lacZ (Fig. 2B; Appendix Fig. S5). This sequence is identical to the

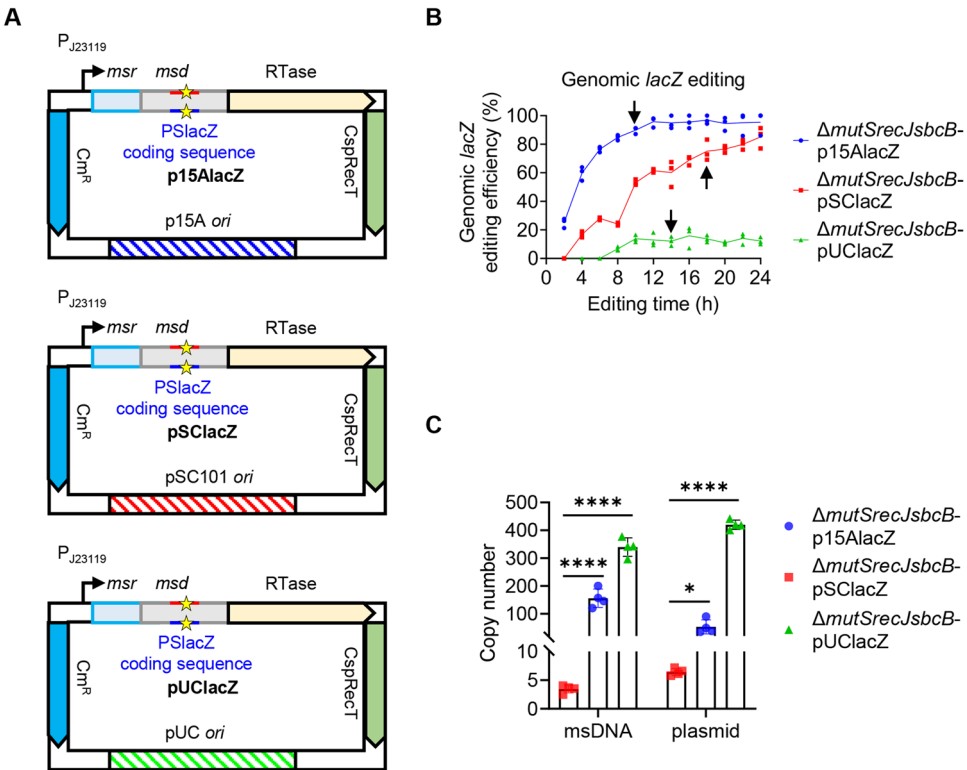

**Figure 1. Effect of plasmid replicons on genomic *lacZ* editing efficiency by the retron system.**

(A) Schematic diagram of plasmids with different replicons. The detailed sequences of the plasmids are provided in Appendix Figs. S2–S4. (B) The efficiency of *lacZ* gene editing was assessed at various time points in Δ*mutSrecJsbcB* strains harboring p15AlacZ, pSClacZ, or pUClacZ plasmids. The data are displayed as a scatter plot, with each data point representing an individual biological replicate (*n* = 3). (C) Plasmid and msDNA copy numbers were quantified in the Δ*mutSrecJsbcB* strain harboring p15AlacZ, pSClacZ, or pUClacZ plasmids at the time points indicated by arrows in panel B (10, 18, and 14 h, respectively). The data were presented as a bar graph, with each bar representing the mean, and error bars indicating the standard deviation. Each data point represents an individual biological replicate (*n* = 4). Statistically significant differences were determined using an unpaired Student's *t*-test. The *p* values for the msDNA copy number comparisons between Δ*mutSrecJsbcB* strains harboring pSClacZ and p15AlacZ, and pSClacZ and pUClacZ were 0.000093 and <0.000001, respectively. For the plasmid copy number comparisons between pSClacZ and p15AlacZ, and pSClacZ and pUClacZ, the *p* values were 0.009991 and <0.000001, respectively. *$P < 0.01$, ****$P < 0.0001$. Source data are available online for this figure.

WT *lacZ* in the genome but differs by 2 bp from the *lacZ* sequence within the retron operon. This introduces another homologous target (Target 3 in Fig. 3B) that can compete with the genomic *lacZ* for binding to msDNA. We continuously monitored the editing efficiency and observed that, starting from 12 h, the editing efficiency of Δ*mutSrecJsbcB*-p15AlacZ-lacZ was significantly lower compared to Δ*mutSrecJsbcB*-p15AlacZ (Fig. 2C). Next, we diluted and plated the Δ*mutSrecJsbcB*-p15AlacZ-lacZ strain cultured for 24 h. Randomly selected colonies were then used for plasmid extraction and sequencing. The sequencing results confirmed that the GG sequence in the *lacZ* region on the plasmid (target 3) mutated to AT (Fig. 2D). This demonstrates that the 92 bp *lacZ* sequence on the plasmid was indeed edited by msDNA, and its presence reduces the efficiency of retron editing, likely due to competition between the plasmid-based sequences and the genomic DNA target locus for binding to the msDNA.

## Verifying msDNA recombination into plasmid *msd* with a dual plasmid system

To investigate msDNA editing the plasmid's *msd* region, we constructed a two-plasmid system. Both plasmids, derived from

pBR and p15A origins, contained the retron operon. We inserted a 92 bp sequence from the kanamycin resistance gene (*kanR*) into the *msd* region of each plasmid. These plasmids, named pBR-PkanX and p15A-PkanY, share high homology (90 bp) within the *kanR* sequence of their *msd* regions, but differ by only two specific bases: T-T in pBR-PkanX and G-A in p15A-PkanY (Fig. 3A; Appendix Figs. S6, S7). We hypothesized that msDNA-kanX generated from pBR-PkanX would serve as a template to edit the *msd* region of p15A-PkanY, specifically mutating the G-A sequence to T-T. Conversely, msDNA-kanY generated from p15A-PkanY would edit the *msd* region of pBR-PkanX, changing the T-T sequence to G-A (Fig. 3B). However, achieving complete editing (100%) of all plasmid copies in a bacterial cell is challenging due to the presence of multiple plasmid copies. Therefore, we anticipate observing double peaks during sequencing at the mutation sites within the *msd* region. This pattern of double peaks would be indicative of successful editing within the *msd* region. One peak would represent the original unedited sequence, while the other peak would represent the incorporated msDNA sequence containing the two-base difference.

The Δ*mutSrecJsbcB* strain was co-transformed with pBR-PkanX and p15A-PkanY plasmids. After a 24-hour incubation to facilitate

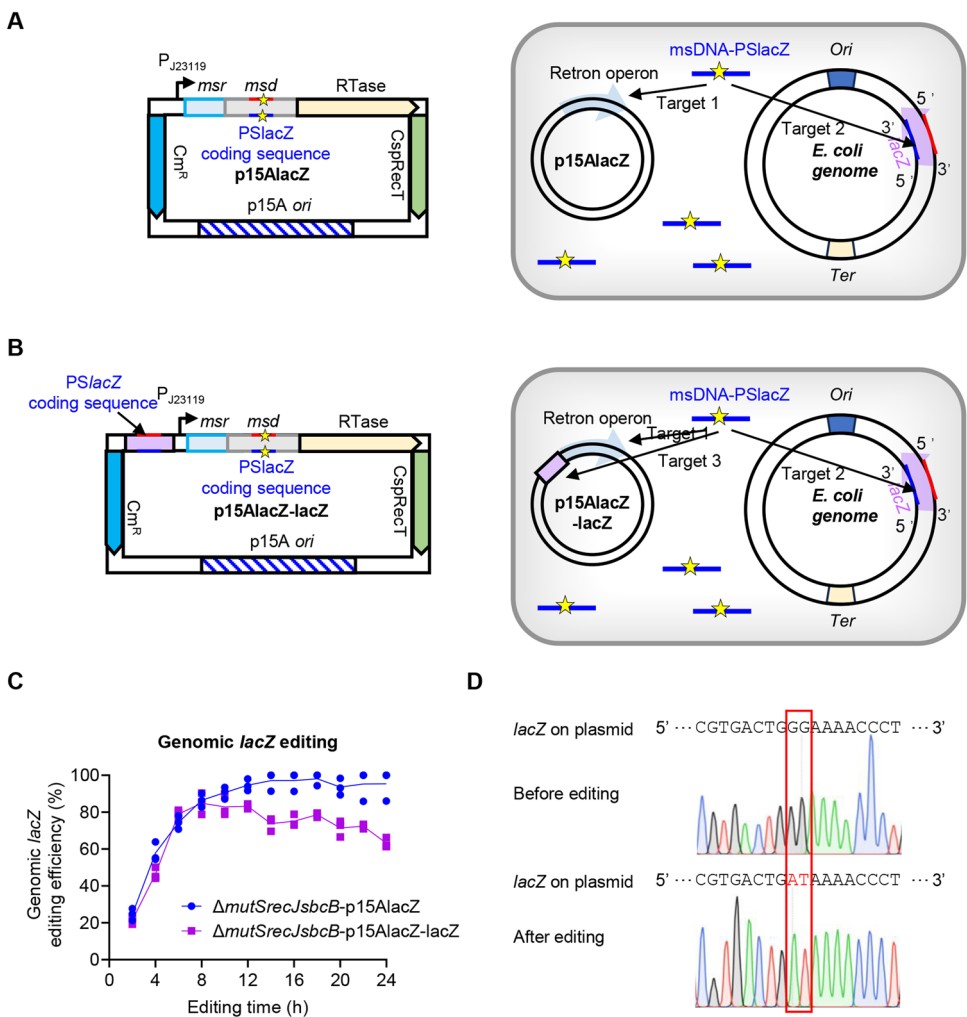

**Figure 2. Additional *lacZ* homology on the plasmid reduces genomic *lacZ* editing efficiency.**

(A) A schematic presentation of the p15AlacZ plasmid. The Δ*mutSrecJsbcB* strain carrying the p15AlacZ plasmid can produce msDNA-PSlacZ, which has the potential to integrate into two targets: the genomic *lacZ* gene and the PSlacZ coding sequence in *msd* on the plasmid. (B) A schematic representation of the p15AlacZ-lacZ plasmid, containing an additional PSlacZ coding sequence. The Δ*mutSrecJsbcB* strain harboring the p15AlacZ-lacZ plasmid can produce msDNA-PSlacZ, which has the potential to integrate into three target sites: the genomic *lacZ* gene, the PSlacZ coding sequence in *msd*, and the additional PSlacZ coding sequence on the plasmid. The detailed sequence of the plasmid is provided in Appendix Fig. S5. (C) Editing efficiency of the *lacZ* gene at different time points in the Δ*mutSrecJsbcB* strain carrying p15AlacZ or p15AlacZ-lacZ plasmids. The data were displayed as a scatter plot, with each data point representing an individual biological replicate ($n = 3$). (D) Sequencing results of the *lacZ* sequence from the target region on plasmid. Red letters represent mutated bases. Source data are available online for this figure.

plasmid replication and *msd* editing, the transformed cells were diluted and spread onto LB agar plates containing streptomycin and chloramphenicol. Only those transformed cells harboring both plasmids were able to survive due to antibiotic resistance. We amplified the *msd* regions of the plasmids by PCR from the colonies of strain 1, using upstream primers located in the sm^R and cm^R gene regions, respectively, along with a downstream primer in the *ret* gene region (see Reagents and tools table). This design ensured that only one plasmid served as the template for PCR amplification. Sequencing of the PCR product revealed a double peak at the 2 bp mutation site, indicating that the *msd* of the plasmid was successfully edited by msDNA (Fig. 3B).

To confirm that *msd* editing originated from the msDNA of the other plasmid, we constructed control plasmids lacking a functional retron promoter. We deleted the retron promoter in pBR-PkanX

and p15A-PkanY, resulting in pBR-ØkanX and p15A-ØkanY plasmids, respectively (Fig. 3A; Appendix Figs. S8, S9). We hypothesized that with an inactive retron operon, *msd* editing would not occur. The Δ*mutSrecJsbcB* strain was co-transformed with three different plasmid combinations: pBR-ØkanX with p15A-PkanY, pBR-PkanX with p15A-ØkanY, and pBR-ØkanX with p15A-ØkanY. This process resulted in the generation of three separate strains, designated as strains 2, 3, and 4, respectively (Fig. 3C). Subsequently, the *msd* regions in all three strains were sequenced to analyze potential editing events. For strain 2, sequencing shows a doublet in the *msd* region of pBR-ØkanX, but a singlet with unaltered bases in p15A-PkanY (Fig. 3C). For strain 3, sequencing shows a singlet with unchanged bases in the *msd* region of pBR-PkanX, but a doublet in p15A-ØkanY (Fig. 3C). For strain 4, sequencing shows a singlet with unchanged bases in

**A**

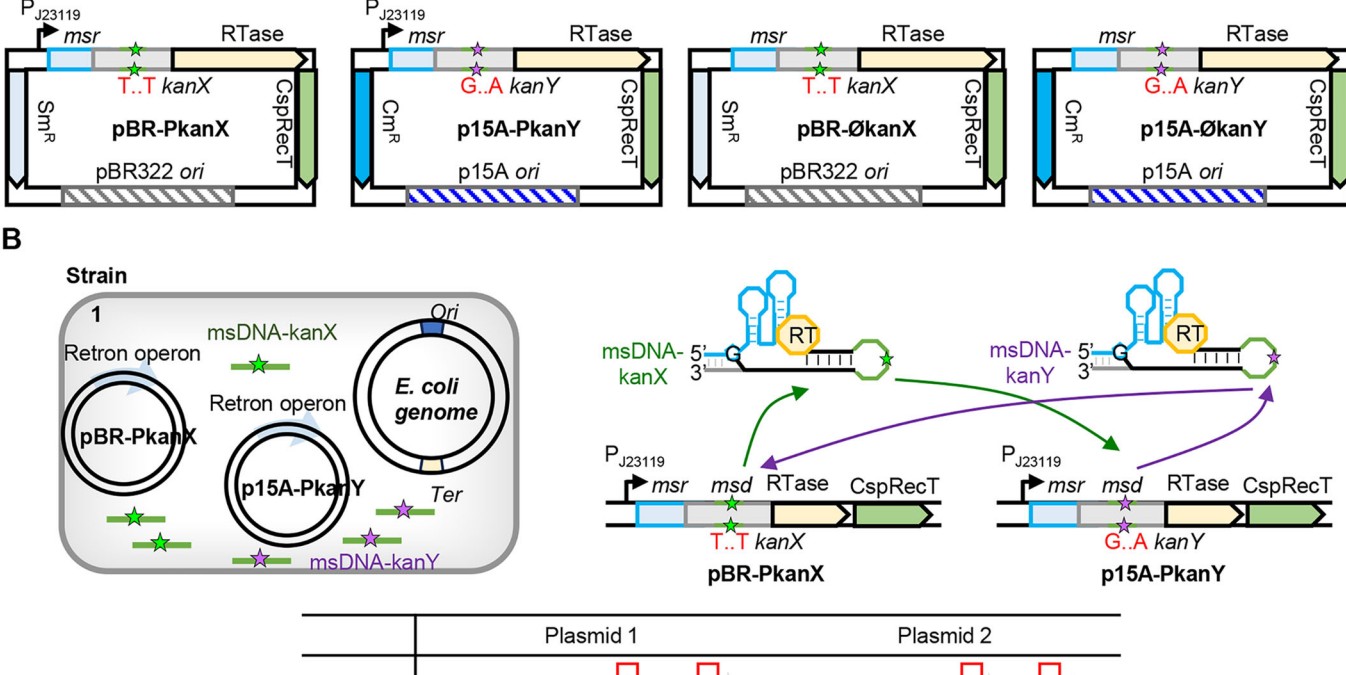

**B**

**Strain**

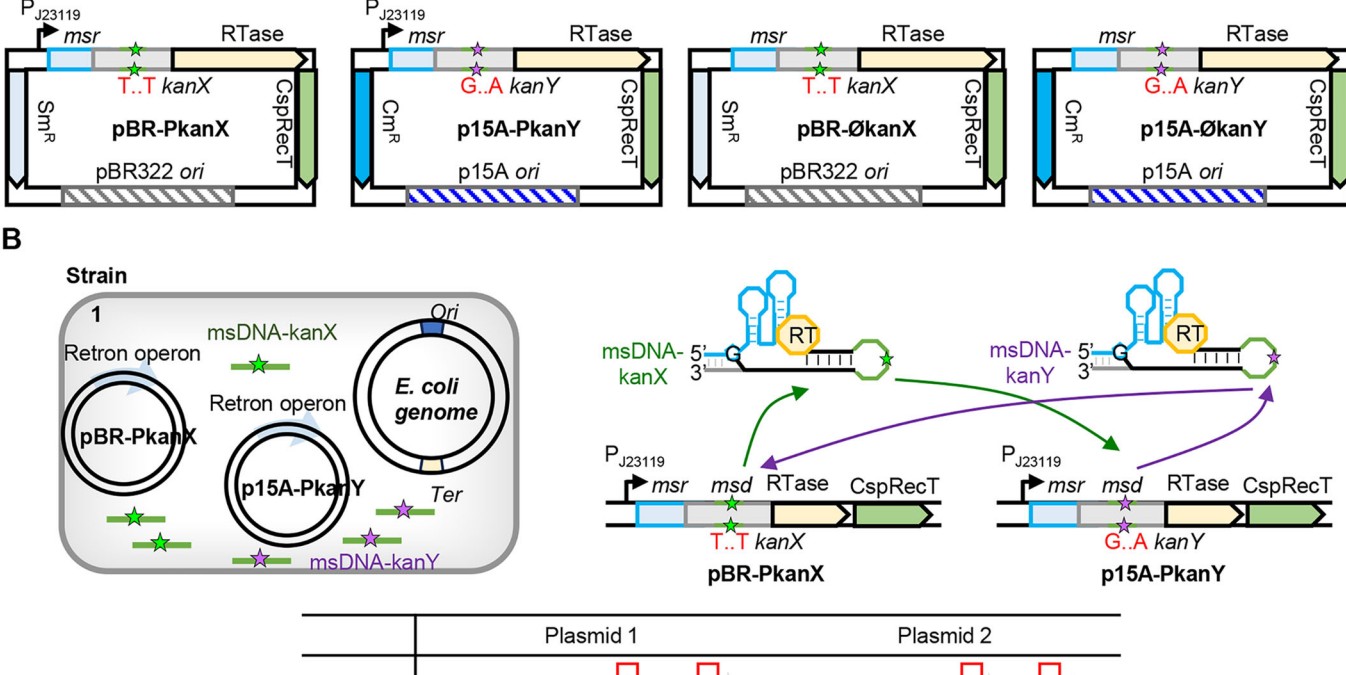

**C**

**Strain 2**

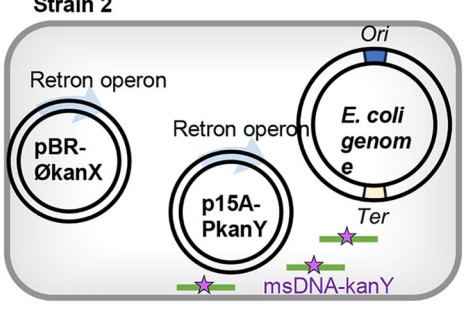

**Strain 3**

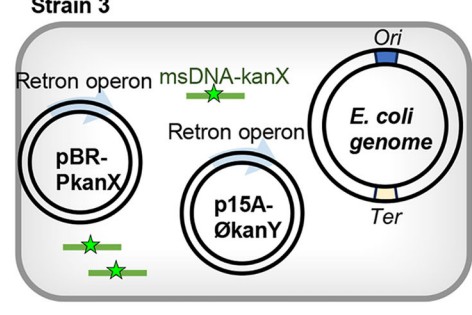

**Strain 4**

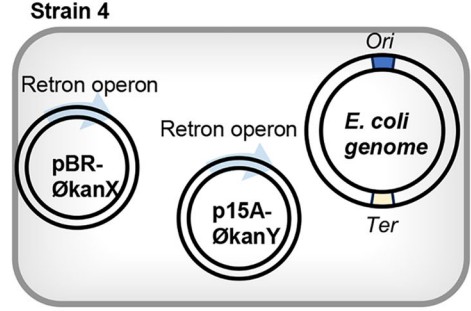

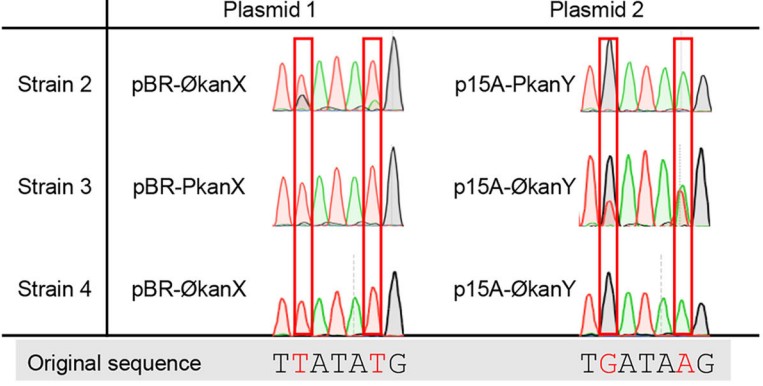

◄

**Figure 3. msDNA-mediated editing of *msd* sequences on plasmids.**

(A) A schematic diagram illustrates four plasmids: pBR-PkanX, p15A-PKanY, pBR-ØkanX, and p15A-ØKanY. Each plasmid contains a 92 bp fragment within its *msd* region homologous to the kanamycin resistance gene (green line). Notably, these homologous sequences share a high degree of similarity (90 bp), differing by only two specific base pairs: "T-T" in pBR-PkanX and "G-A" in pBR-PkanY. Green and purple stars indicate mutation sites. Plasmids pBR-PkanX and p15A-PKanY possess the J23119 promoter, enabling retron operon transcription and subsequent msDNA expression. In contrast, the J23119 promoter is absent in pBR-ØkanX and p15A-ØKanY, preventing retron operon transcription and msDNA production. The detailed sequences of the plasmids are provided in Appendix Figs. S6–S9. (B) The Δ*mutSrecJsbcB* strain harboring two plasmids, pBR-PkanX and p15A-PKanY (strain 1), was constructed. The msDNA produced by each plasmid can edit the homologous sequence within the *msd* gene of the other plasmid. Sequencing results of the targeted *msd* gene region are provided. (C) Schematic diagrams illustrate strains 2, 3, and 4, each carrying different plasmid combinations, as shown in the figure. Sequencing data of the targeted *msd* gene regions from these strains reveal mutations introduced through editing, denoted by red letters.

the *msd* regions of both plasmids (Fig. 3C). The above results demonstrate that msDNA synthesized by an active retron operon can edit *msd* on the plasmid.

## Effects of *msd* positioning and enhanced retron system on editing efficiency

Because the homologous sequence in msDNA has a stronger editing ability when it is consistent with the lagging strand sequence of the replication fork (Ellis et al, 2001; Gallagher et al, 2014; Huen et al, 2006), we hypothesized that msDNA-PSlacZ could act as both the lagging strand during genomic DNA replication and pUClacZ plasmid replication. This suggests that msDNA-PSlacZ might be intensively used for editing of the pUC plasmid's *msd*, depleting the pool of msDNA available for editing the genomic *lacZ* target, and thus reducing its editing efficiency (Fig. 4A).

To address this potential competition, we took advantage of the fixed replication start site (*ori*) and termination site (*ter*) in the pUC plasmid. We strategically relocated the *msd* to a different position on the plasmid, across the *ter* boundary. This relocation would alter the role of msDNA-PSlacZ during replication. Instead of acting as the lagging strand, msDNA-PSlacZ would act as the leading strand, preventing it from editing the *msd* once it moves across the *ter* boundary. The pUC plasmid, which replicates via the ColE1-based *ori* in a theta (θ) fashion (Lilly and Camps, 2015), has an unclear exact *ter* location, though it is believed that ColE1 plasmids replicate at different rates on either side of the *ori* (Lovett et al, 1974; Viguera et al, 1996). Based on this uncertainty, we relocated the *msd* position on the plasmid clockwise by 991 bp from its original position while keeping the plasmid size unchanged, resulting in the creation of the pUClacZ1 plasmid (Fig. 4B; Appendix Fig. S10). Compared to Δ*mutSrecJsbcB*-pUClacZ, the editing efficiency of Δ*mutSrecJsbcB*-pUClacZ1 increased from 5% at 2 h to ~70% after 24 h (Fig. 4C), demonstrating the efficacy of *msd* relocation for enhanced editing efficiency.

To further increase this efficiency, we repositioned the *msd* an additional 827 bp clockwise, creating the pUClacZ2 plasmid (Fig. 4D; Appendix Fig. S11). This adjustment significantly enhanced genomic *lacZ* editing efficiency, with the Δ*mutSrecJsbcB*-pUClacZ2 achieving over 80% editing within just 2 h and nearly 100% after 6 h (Fig. 4E). But, after 18 h, the efficiency temporarily dropped to 70% before recovering to nearly 100% by 22 h (Fig. 4E). The substantial increase in editing efficiency in the Δ*mutSrecJsbcB*-pUClacZ2 strain suggests that the *ter* site lies within the 827 bp range and that this repositioning shifted the msDNA-PSlacZ from the lagging to the leading strand. This reduced *msd*

editing on the plasmid while enhancing genomic *lacZ* editing. In addition, we observed a significant decrease in the growth rate of the Δ*mutSrecJsbcB*-pUClacZ2 strain (Fig. 4E). We hypothesized that this decrease may be due to reduced plasmid stability, leading to a significant loss of plasmids. This loss could result in a sharp decrease in the number of antibiotic-resistant colonies, contributing to the observed decrease in growth rate.

To address these issues, we constructed the pUClacZ3 plasmid by reversing the orientation of the *ori* (Fig. 4F; Appendix Fig. S12). Given the unequal replication rates on either side of the *ori*, we hypothesized that this reversal would shift the *ter* position by 180 degrees, transitioning msDNA from the lagging strand to the leading strand. As anticipated, Δ*mutSrecJsbcB*-pUClacZ3 maintained high genomic editing efficiency and excellent stability without negatively impacting bacterial growth (Fig. 4G). These findings underscore the importance of minimizing *msd* editing to enhance genomic editing efficiency. Our experiments demonstrate that reducing *msd* editing is a powerful strategy for improving gene editing outcomes.

To demonstrate the versatility of the improved retron system, we applied it to new targets in the *E. coli* genome. Two genes, *ung* (involved in DNA repair) and *betI* (choline metabolism), located on opposite sides of the genomic *ori*, were chosen for testing. We replaced the *lacZ* homologous sequence in pUClacZ3 with the corresponding homologous sequences of *ung* and *betI* (Appendix Figs. S13, S14), using retron editing to introduce two-point mutation in *ung* and one base pair deletion in *betI* (Fig. EV2A,B). Sequencing results revealed successful editing for both genes within 2 h of liquid culture (Fig. EV2C). This demonstrates the broad applicability of our system for targeting various genome genes for editing.

In this study, we investigated how varying the number of msDNA template sequences (*msd*) affects genome editing efficiency in *E. coli*, a model prokaryotic strain. While increasing the copy number of *msd* enhanced msDNA expression, it also led to unwanted competition, ultimately reducing editing efficiency. In conventional ssDNA-mediated gene editing, ssDNA is introduced into the cell via electroporation without requiring intracellular template synthesis. In contrast, retron editing synthesizes ssDNA within the cell using *msd* as a template. This introduces competition, which is absent in traditional ssDNA methods. Despite increasing interest in retron technology, prior studies have overlooked this competition issue. Our study is the first to identify and confirm *msd* editing by msDNA, and we propose a novel method to reduce this competition, significantly enhancing target gene editing efficiency.

Retrons have gained significant interest for their potential as powerful in vivo DNA writing tools, exemplified by SCRIBE

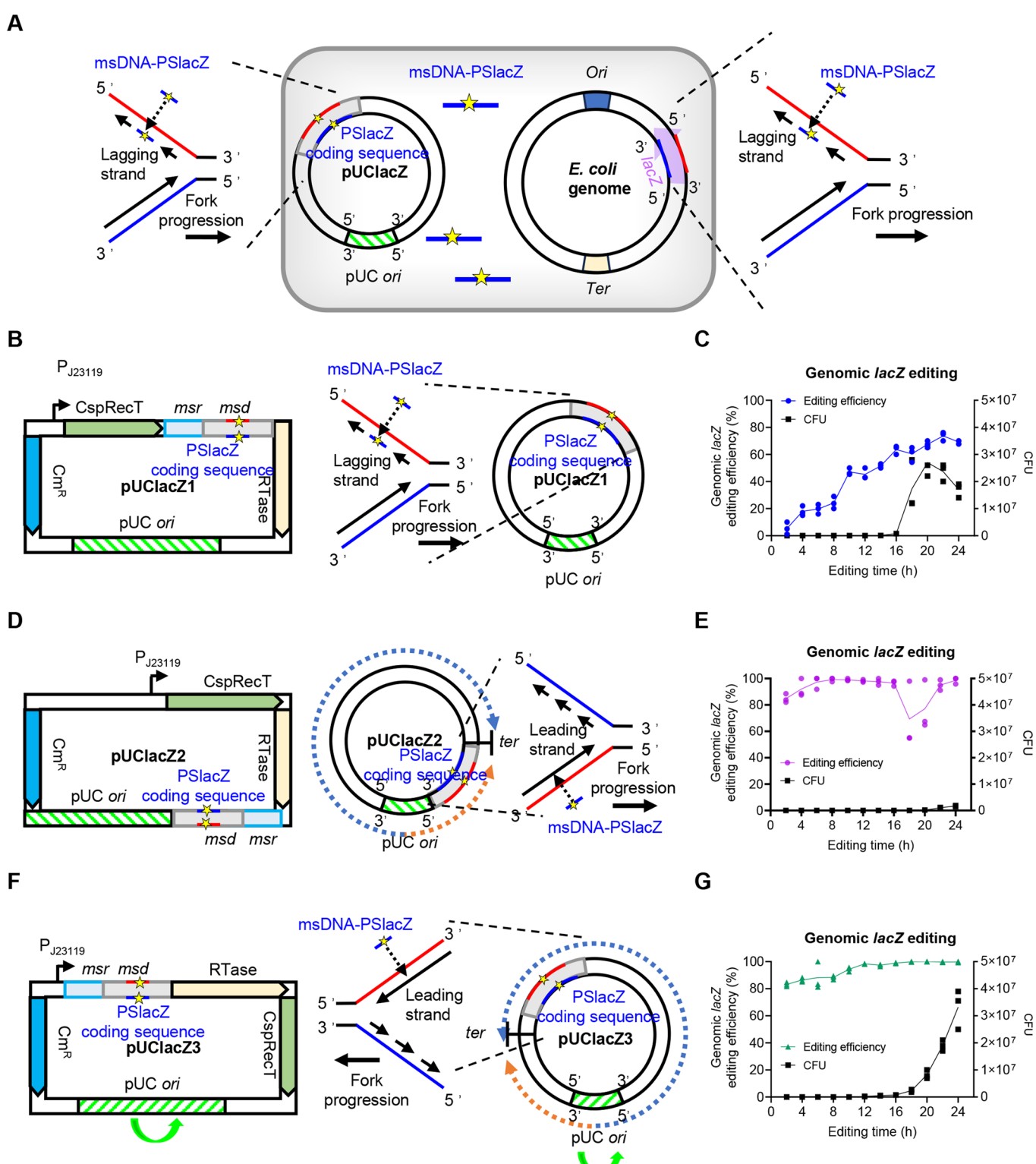

(Synthetic Cellular Recorders Integrating Biological Events) (Farzadfard and Lu, 2014; Lear et al, 2023). However, a major challenge for SCRIBE is its slow writing speed. Only one in 10,000 *E. coli* cells integrates the desired edit per generation (Farzadfard and Lu, 2014), and even the improved HiSCRIBE version requires ~60 generations (2 days) for complete writing (Farzadfard et al,

2021). This slow writing speed arises from the inefficiency of the editing process. Because relatively few cells successfully incorporate the desired mutation in each generation, many cell divisions are required before the edited population becomes a significant proportion of the overall culture. This limitation hinders SCRIBE's ability to record rapid environmental changes. Theoretically, with

◀

**Figure 4.   Genomic *lacZ* gene editing in Δ*mutSrecJsbcB* strains harboring pUClacZ1, pUClacZ2, or pUClacZ3 plasmids.**

(A) In *E. coli* strain carrying the pUClacZ plasmid, msDNA-PSlacZ shares the same sequence as the lagging strand in both the plasmid and genomic DNA replication forks. This enables msDNA integration into both plasmid *msd* and genomic *lacZ*. (B) Schematic diagram of the pUClacZ1 plasmid. Left: The overall structure of the pUClacZ1 plasmid. Note that the *msd* has been relocated in pUClacZ1. msDNA-PSlacZ integrates into the plasmid replication fork as the lagging strand due to its sequence complementarity with the red line sequence. This lagging-strand integration enables efficient *msd* editing. The detailed sequence of the plasmid is provided in Appendix Fig. S10. (C) Editing efficiency and growth in a colony-forming unit (CFU) of Δ*mutSrecJsbcB*-pUClacZ1 strain at different time points. The data were displayed as a scatter plot, with each data point representing an individual biological replicate (n = 3). (D) Schematic diagram of the pUClacZ2 plasmid. Left: The overall structure of the pUClacZ2 plasmid. Note that the *msd* has been relocated in pUClacZ2. The blue and orange dashed arrows represent the regions where the replication forks move on the left and right sides of the plasmid, respectively. The T-shaped symbol indicates the replication termination (*Ter*) site. msDNA-PSlacZ integrates into the plasmid replication fork as the leading strand. This leading-strand integration hinders efficient *msd* editing. The detailed sequence of the plasmid is provided in Appendix Fig. S11. (E) Editing efficiency and growth in CFU of Δ*mutSrecJsbcB*-pUClacZ2 strain at different time points. The data were displayed as a scatter plot, with each data point representing an individual biological replicate (n = 3). (F) Schematic diagram of the pUClacZ3 plasmid. Left: The overall structure of the pUClacZ3 plasmid. Note that the replication origin (*ori*) has been flipped horizontally, which in turn swaps the plasmid's left and right replication forks. This reorientation shifts the replication terminus (*ter*) to the opposite end of the plasmid and positions msDNA-PSlacZ into the leading strand of the plasmid replication fork. This leading-strand integration hinders efficient *msd* editing. The detailed sequence of the plasmid is provided in Appendix Fig. S12. (G) Editing efficiency and growth in CFU of Δ*mutSrecJsbcB*-pUClacZ3 strain at different time points. The data were displayed as a scatter plot, with each data point representing an individual biological replicate (n = 3). Source data are available online for this figure.

100% efficiency of msDNA integration at the replication fork, after one round of replication, the DNA would form 2 double strands, with no pure mutants yet. Following a second round, there would be 4 double strands and 1 pure mutant. By extension, after six rounds, there would be 64 double strands and 57 pure mutants. In other words, the theoretical maximum mutation rate after six rounds of replication (~6 generations) is approximately 89%. In our work, we were able to achieve over 80% efficiency in editing bacteria within ~6 generations (2 h). It suggests that the method is approaching its theoretical limit. This dramatic increase in writing speed paves the way for real-time recording of dynamic environmental changes using retron systems, opening exciting possibilities for environmental monitoring and synthetic biology applications.

In applying these conclusions to plasmids with various replication origins, it's crucial to note that *msd*-induced competition is more significant in high-copy plasmids compared to medium- and low-copy ones. To reduce this competition in high copy number plasmids, we propose two strategies: (1) adjusting the relative position of *msd* on the plasmid and (2) reversing the orientation of the plasmid replicon. For plasmids with a theta replication mechanism that replicates at the same rate in both directions, the first strategy is effective, while the second is not. However, the first approach may carry potential risks, such as altered msDNA production or reduced plasmid stability. In contrast, both strategies are effective for plasmids that replicate at different rates in each direction. In these cases, we prefer the second method, as it reduces competition while minimizing the impact on plasmid stability and preserving the operon's function.

The successful application of the retron editing system in various organisms, including yeast (Gonzalez-Delgado et al, 2024; Sharon et al, 2018), mammalian cells (Kong et al, 2021; Lopez et al, 2022; Zhao et al, 2022), and plants (Jiang et al, 2022), has highlighted a potential issue: the competition between *msd* and genomic DNA targets within eukaryotic cells. Unlike bacteria, eukaryotes utilize the retron system in conjunction with CRISPR-Cas9 tools, which create double-strand breaks (DSBs) in genomic DNA. Here, msDNA plays a crucial role in repairing DSBs by providing a template for DNA polymerase to fill the gap and integrate the desired edits, a process known as homology-directed repair (HDR) (Sharon et al, 2018; Zhao et al, 2022). To express msDNA in eukaryotic systems, retrons are typically integrated into shuttle plasmids, viral vectors, or directly into the host genome,

resulting in multiple or single copies of *msd* within the cell. Studies in yeast have shown that ssDNA can participate in DNA replication as the lagging strand during plasmid replication, similar to the process observed in prokaryotes (Barbieri et al, 2017). This suggests that msDNA could potentially target originating plasmids or viral vectors and participate in their replication process. Such self-targeting may reduce the pool of msDNA available for effective HDR, thereby decreasing the overall efficiency of the editing process. Supporting this notion, a study by Zhao et al observed a discrepancy between msDNA abundance and editing efficiency in different retron systems. While Retron Sa163 from *Stigmatella aurantiaca* generated the most msDNA, it exhibited a lower HDR rate compared to Eco1 (Zhao et al, 2022). Furthermore, certain retron editing systems that demonstrate high efficiency in prokaryotes do not yield similar results in eukaryotic cells (Khan et al, 2024; Lopez et al, 2022). We believe this difference arises from variations in editing mechanisms. In prokaryotes, msDNA is utilized during DNA replication for editing, whereas in eukaryotes, it serves as a template for repair during the HDR process following DSBs. Therefore, in addition to the competitive mechanisms revealed in this study, we speculate that another significant factor limiting the efficiency of retron editing in eukaryotes is the inefficiency of HDR.

## Methods

### Reagents and tools table

| Reagent/resource | Reference or source | Identifier or catalog number |
|---|---|---|
| **Experimental models** | | |
| *E. coli* DH5α | ThermoFisher | 18265017 |
| *E. coli* MG1655 WT | ATCC | 700926 |
| *E. coli* Δ*mutS* | This study | N/A |
| *E. coli* Δ*mutSrecJ* | This study | N/A |
| *E. coli* Δ*mutSrecJsbcB* | This study | N/A |
| **Recombinant DNA** | | |
| pFF745 | Farzadfard and Lu, 2014 | Addgene #61450 |

| Reagent/resource | Reference or source | Identifier or catalog number |
|---|---|---|
| pCas | Jiang et al, 2015 | Addgene #62225 |
| pTargetF | Jiang et al, 2015 | Addgene #62226 |
| pWYF136 | This study | N/A |
| pWYF154 | This study | N/A |
| pWYF153 | This study | N/A |
| p15APSlacZ-anti see Appendix Fig. S1 | This study | N/A |
| p15APSlacZ see Appendix Fig. S2 | This study | N/A |
| pSClacZ see Appendix Fig. S3 | This study | N/A |
| pUClacZ see Appendix Fig. S4 | This study | N/A |
| p15AlacZ-lacZ see Appendix Fig. S5 | This study | N/A |
| pBR-PkanX see Appendix Fig. S6 | This study | N/A |
| p15A-PkanY see Appendix Fig. S7 | This study | N/A |
| pBR-ØkanX see Appendix Fig. S8 | This study | N/A |
| p15A-ØkanY see Appendix Fig. S9 | This study | N/A |
| pUClacZ1 see Appendix Fig. S10 | This study | N/A |
| pUClacZ2 see Appendix Fig. S11 | This study | N/A |
| pUClacZ3 see Appendix Fig. S12 | This study | N/A |
| pUCung see Appendix Fig. S13 | This study | N/A |
| pUCbetI see Appendix Fig. S14 | This study | N/A |
| **Oligonucleotides and other sequence-based reagents** | | |
| 136-F | This study | ATCTATTACCCTGT TATCCCCACCACCGAC |
| 136-R | This study | GCCATCGGGATCGGCTCT CCACTAGTATTATACCTAGGAC |
| mut-s-F | This study | GGAGAGCCGATCCCGATGGCG TTTTAGAGCTAGAAATAGC |
| mut-s-R | This study | CAAGTACGCAAAATTGTATCCA GGTCGACTCTAGAGAATT |
| mut-ty1-F | This study | GATACAATTTTGCGTACTTGC |
| mut-ty1-R | This study | AGACTATCGGGAATTGTTA GGGGTTATGTCCGGTTCCCT |
| mut-ty2-F | This study | TAACAATTCCCGATAGTCTT TTGCTATCGGG |
| mut-ty2-R | This study | GGGGATAACAGGGTAATAGATGA TTGTAATAACTGATATTC |
| 154-F | This study | ATCTATTACCCTGTTATCCCC ACCACCGAC |

| Reagent/resource | Reference or source | Identifier or catalog number |
|---|---|---|
| 154-R | This study | ATGACATTGAATTCGCTATCACTA GTATTATACCTAGGAC |
| sbc-s-F | This study | GATAGCGAATTCAATGTCATGTT TTAGAGCTAGAAATAGC |
| sbc-s-R | This study | CATTCCTTTTTCACCTCAGCCCA GGTCGACTCTAGAGAATTC |
| sbc-ty1-F | This study | GGCTGAGGTGAAAAAGGAATG |
| sbc-ty1-R | This study | CCAGCGGCGGAGGCTTCAAAT AGGTTAAATCCGTTATTTC |
| sbc-ty2-F | This study | TTTGAAGCCTCCGCCGCTG GTACGG |
| sbc-ty2-R | This study | GGGATAACAGGGTAATAGA TCTGCTGCGGCAAATTAAGT AAGG |
| 153-F | This study | TCAATGACATCAACAGAGGCA CTAGTATTATACCTAGGAC |
| 153-R | This study | GGCTGGCGTATAAACGGCGCA CTAGTATTATACCTAGGAC |
| recJ-s-F | This study | GCGCCGTTTATACGCCAGCCG TTTTAGAGCTAGAAATAGC |
| recJ-s-R | This study | GGCGCGGGAAAGCAAGATAACA GGTCGACTCTAGAGAATT |
| recJ-ty1-F | This study | TTATCTTGCTTTCCCGCGCC |
| recJ-ty1-R | This study | TTTTATAGAGAAGATGACGG CGAATTATTTACCGCTGGTC |
| recJ-ty2-F | This study | CGTCATCTTCTCTATAAAAA AGAGCGTGG |
| recJ-ty2-R | This study | GGGATAACAGGGTAATAGATA GGGTGTCGACAACGGCTTC |
| ori-p15-F | This study | AAGGCCGCGTTGCTGGCGTT AGAATATGTGATACAGGATA |
| ori-p15-R | This study | ATCTGGCGAAAATGAGACGTTG ATCGGCACGTAAGAGGTTC |
| R-J19-F | This study | TAGCTCAGTCCTAGGTATAATA CTAGTATCAGCAGGACGC ACTGACC |
| R-J19-R | This study | TATACCTAGGACTGAGCTAGCT GTCAATTGCTAGCATCTC GAGGTG |
| msr-F | This study | AATTCAGGAAACCC GTTTTTTCTGAC |
| msr-R | This study | GAATTCAGGAAAACAGACA GTAACT |
| Z2-F | This study | TGTCTGTTTTCCTGAATTCGG GGATGTGCTGCAAGGCGA |
| Z2-R | This study | AAACGGGTTTCCTGAATTCC ATGATTACGGATTCACTGGCC |
| TBZ-F | This study | CGTGACTGATAAAACCCTGGCGT TACCCAACTTAATC |
| TBZ-R | This study | AGGGTTTTATCAGTCACGACGT TGTAAAACG |
| CspRecT-F | This study | ATGAACCAAATCGTGAA GTTCACTG |
| CspRecT-R | This study | TTAAAACACCTCGCCCT CGTCAAC |

| Reagent/resource | Reference or source | Identifier or catalog number |
|---|---|---|
| 187-F | This study | CGAGGGCGAGGTGTTTTAAACGCGTGCTAGAGGCATC |
| 187-R | This study | CAGTGAACTTCACGATTTGGTTCATTCGTTTTATACCTCTGAATC |
| Z1-F | This study | TGTCTGTTTTCCTGAATTCCCATGATTACGGATTCACTGGCC |
| Z1-R | This study | AAACGGGTTTCCTGAATTGGGGATGTGCTGCAAGGCGA |
| msr-F | This study | AATTCAGGAAACCCGTTTTTTCTGAC |
| msr-R | This study | GAATTCAGGAAAACAGACAGTAACT |
| pSC101-F | This study | GAAAAGATCAAAGGATCTTCCGTTTTCATCTGTGCATATGGAC |
| pSC101-R | This study | GTGTGAAATACCGCACAGATGGTGCTACTTAAGCCTTTAG |
| 240-F | This study | ATCTGTGCGGTATTTCACACCGCTCAGTGGAACGAAAACTC |
| 240-R | This study | GAAGATCCTTTGATCTTTTCCTCGCCGCAGCCGAACGCCT |
| ori-pUC-F | This study | AAGGCCGCGTTGCTGGCGTTTTTCCATAGGCTCCGCCCCC |
| ori-pUC-R | This study | ATCTGGCGAAAATGAGACGTTGATCGGCACGTAAGAGGTTC |
| 239-F | This study | ACGTCTCATTTTCGCCAGATATCG |
| 239-R | This study | AACGCCAGCAACGCGGCC |
| 41-lacZ-F | This study | CGGTATTTCACACCGCATAGGGGATGTGCTGCAAGGCGAT |
| 41-lacZ-R | This study | GACATTGCACTCCACCGCTGACCATGATTACGGATTCACTGGCCG |
| 41-F | This study | TCAGCGGTGGAGTGCAATGTCTCGTCTTCACCTCGAGATGCTAGC |
| 41-R | This study | TATGCGGTGTGAAATACCGAAGGGCCTCGTGATACGCCTATT |
| kan-F | This study | TCTGTTTTCCTGAATTCGTCTTGCTCTAGGCCGCGA |
| kan-R | This study | AAACGGGTTTCCTGAATTCGTCGCACCTGATTGCCCGA |
| msr-F | This study | AATTCAGGAAACCCGTTTTTTCTGAC |
| msr-R | This study | GAATTCAGGAAAACAGACAGTAACT |
| 187-F | This study | CGAGGGCGAGGTGTTTTAAACGCGTGCTAGAGGCATC |
| 745-R | This study | AAGGTAGTCGGCAAATAATTTGATATCGAGCTCGCTTGG |
| Sep-F | This study | TTATTTGCCGACTACCTTGGTGATCTCGCC |
| Spe-R | This study | ATGAGGGAAGCGGTGATCGCC |
| CatP-F | This study | TCACCGCTTCCCTCATTTTAGCTTCCTTAGCTCCTGAAAATCTCG |

| Reagent/resource | Reference or source | Identifier or catalog number |
|---|---|---|
| CspRecT-R | This study | TTAAAACACCTCGCCCTCGTCAAC |
| kan-F | This study | TCTGTTTTCCTGAATTCGTCTTGCTCTAGGCCGCGA |
| kan-R | This study | AAACGGGTTTCCTGAATTCGTCGCACCTGATTGCCCGA |
| msr-F | This study | AATTCAGGAAACCCGTTTTTTCTGAC |
| msr-R | This study | GAATTCAGGAAAACAGACAGTAACT |
| GA-F | This study | GCTGATTGATAAGGGTATAAATGGGCTCGCG |
| GA-R | This study | ATACCCTTATCAATCAGCATCCATGTTGGA |
| 187-F | This study | CGAGGGCGAGGTGTTTTAAACGCGTGCTAGAGGCATC |
| CspRecT-R | This study | TTAAAACACCTCGCCCTCGTCAAC |
| dP-F | This study | ATGCTAGCAAATCAGCAGGACGCACTGACC |
| dP-R | This study | TGCTGATTTGCTAGCATCTCGAGGTG |
| RT-rrnB-F | This study | CCTAAGGATCCGGTTGATAACGCGTGCTAGAGGCATCAAATAAAACGA |
| RT-R | This study | ATCAACCGGATCCTTAGGTCTTC |
| P-RecT-F | This study | TTGATTCAGAGGTATAAAACGAATGAACCA |
| CspRecT-R | This study | TTAAAACACCTCGCCCTCGTCAAC |
| RecT-msr-F | This study | GAGGGCGAGGTGTTTTAATCAGCAGGACGCACTGACCTT |
| RecT-P-R | This study | TTTTATACCTCTGAATCAATACTAGTATTATACCTAGGACTGAGCTAGC |
| P-RecT-F | This study | TTGATTCAGAGGTATAAAACGAATGAACCA |
| RT-RecT-R | This study | GATTTCATGAAAGTTTAAAACACCTCGCCCTCGTCAAC |
| RecT-RT-F | This study | AGGGCGAGGTGTTTTAAACTTTCATGAAATCCGCTGAATATTTGAACAC |
| msr-RT-R | This study | TATCAACCGGATCCTTAGGTCTTCGCTT |
| RT-msr-F | This study | TAAGGATCCGGTTGATATCAGCAGGACGCACTGACCT |
| rrnB-msr-R | This study | ATGCCTCTAGCACGCGTTGCGCACCCTTACGTCAGAA |
| rrnB-T1-F | This study | ACGCGTGCTAGAGGCATCAAATA |
| ori-F | This study | CCCTTAACGTGAGTTTTCGTTCCAC |
| ori-R | This study | CCTAGACCTAGGCGTTCGGCTGC |
| 239NY-F | This study | AACGCCTAGGTCTAGGTCATGACTAGTGCTTGGATTCTC |

| Reagent/resource | Reference or source | Identifier or catalog number |
|---|---|---|
| 239NY-R | This study | GAAAACTCACGTTAAGGGCGGC GGATTTGTCCTACTCA |
| pBR-F | This study | ACTTCGGCGATCACCGCTTC |
| ret-R | This study | CGAGATTTCTCAATCTAAAAG |
| p15A-F | This study | CATTGGGATATATCAACGGT |
| ret-qPCR-F | This study | AGGCGACTCGCATATCTGTTG |
| ret-qPCR-R | This study | CGTAGAACCCATCCTTGTAAGGC |
| msd-qPCR-F | This study | CTGAGTTACTGTCTGTTTTCCTG |
| msd-qPCR-R | This study | TCAGAAAAAACGGGTTT CCTGAATTC |
| tdk-qPCR-F | This study | CCGCAATGAATGCGGGGTAAG |
| tdk-qPCR-R | This study | GCAGGCGATGACAAACCTATACG |
| **Chemicals, Enzymes and other reagents** | | |
| Yeast Extract | Angelyeast | FM888 |
| Tryptone | Angelyeast | FP318 |
| Sodium chloride | Sinopharm | 10019308 |
| Agar powder | Bomeibio | BBM7064-2.5KG |
| Glycerol | Sinopharm | 10010618 |
| Isopropyl β-D-1-thiogalactopyranoside | Biofroxx | 1122GR005 |
| X-gal | Biofroxx | 1100MG100 |
| Calcium chloride | Sinopharm | 10005817 |
| Chloramphenicol | Solarbio | C8050 |
| Kanamycin | Solarbio | K8020 |
| **Software** | | |
| GraphPad Prism version 10.2.0 | https://www.graphpad.com | |
| SnapGene 7.2 | https://www.snapgene.com | |
| **Other** | | |
| 2 × Phanta Max Master Mix (Dye Plus) | Vazyme | P525-AA |
| Hieff Clone® Plus One Step Cloning Kit | Yeasen | 10911ES |
| Hieff® qPCR SYBR Green Master Mix (Low Rox Plus) | Yeasen | 11202ES |
| AFTSpin Plasmid Mini Kit | Abclonal | RK30101 |

## Plasmid construction

We constructed plasmids using pFF745 as the parent vector (Farzadfard and Lu, 2014). Primer pairs used for plasmid construction were synthesized from Tianyi Huiyuan (Wuhan, Hubei, China) (see Reagents and tools table). All plasmids generated in this study were assembled using the Hieff Clone Plus multi-one-step cloning kit (Yeasen, Shanghai, China) and confirmed by sequencing (Quintarabio, Wuhan, Hubei, China). The constructed plasmids were transformed into bacterial cells using the calcium chloride (CaCl₂) method to generate corresponding derivative strains. Plasmids and their sequence are shown in the Appendix.

## Bacterial strains and growth conditions

*E. coli* DH5a was used for all cloning experiments. Δ*mutS*, Δ*mutSrecJ*, and Δ*mutSrecJsbcB* strains were generated via the CRISPR-Cas9 System (Jiang et al, 2015). All strains were grown in LB medium (containing 10 g tryptone, 5 g yeast extract, and 10 g NaCl per liter) at 37 °C supplemented with chloramphenicol (15 µg/mL) or spectinomycin (20 µg/mL) when necessary.

## Editing efficiency quantification

We used blue-white screening to quantify gene editing efficiency. Approximately 200 ng of the plasmid of interest was added to 100 µL of competent Δ*mutSrecJsbcB* cells. The mixture was incubated on ice for 30 min, followed by a heat shock at 42 °C for 90 s. Afterward, the cells were placed on ice for an additional 2 min. To promote cell recovery, 900 µL of pre-warmed LB medium was added, and the suspension was incubated at 37 °C with shaking (200 rpm) for one hour. Following recovery, 1 mL of culture was transferred to fresh LB medium (50 mL) containing chloramphenicol (50 µg/mL) and incubated at 37 °C with shaking (200 rpm) until reaching the desired time point. The culture was then diluted and plated onto LB agar containing IPTG, X-gal, and chloramphenicol. After overnight incubation at 37 °C, white and blue colonies were counted, and editing efficiency was calculated as the percentage of white colonies relative to the total number of colonies.

We categorized the bacteria into three groups based on their editing status: Group 1 consists of fully edited bacteria, which are completely edited in the liquid medium, resulting in colonies that contain 100% non-functional *lacZ* and appear as white colonies. Group 2 includes partially edited bacteria that are unedited in the liquid medium but undergo editing while grown on a solid medium, causing these colonies to exhibit a slight blue color due to the presence of unedited bacteria. Group 3 comprises unedited bacteria, which retain functional *lacZ* and thus form blue colonies due to X-gal hydrolysis. In our analysis, we recognized only the first group of bacteria as edited.

## Determination of plasmid and msDNA copy numbers

To estimate plasmid and msDNA copy numbers, we employed Real-time quantitative PCR (qRT-PCR) following a cell lysis step. Briefly, 100 µL of logarithmically growing cells were lysed by boiling for 10 min. Lysates containing genomic DNA, plasmid DNA, and msDNA were stored at −20 °C until further analysis (Lopez et al, 2022).

Three sets of primers were used for qRT-PCR: (1) tdk-qPCR primers: These primers (tdk-qPCR-F and tdk-qPCR-R) amplify genomic DNA and generate a cycle threshold (Ct) value called $C_{tg}$. (2) ret-qPCR primers: These primers (ret-qPCR-F and ret-qPCR-R) amplify plasmid DNA, producing a Ct value named $C_{tp}$. (3) msd-qPCR primers: This primer set (msd-qPCR-F and msd-qPCR-R) targets the msr/msd region, amplifying the corresponding sequences on msDNA and plasmids. This amplification generates a Ct value called $C_{tm}$.

The plasmid and msDNA copy numbers were estimated using qRT-PCR data by comparing the Ct values from genomic, plasmid,

and msDNA sequences. The formulas used account for the differences in amplification of these DNA types.

The plasmid copy number was estimated using the formula:

$$\text{plasmid copy number} = 2^{-\Delta Ct} = 2^{(Ctg-Ctp)}$$

This formula estimates how many more plasmid copies exist relative to the genomic DNA.

The msDNA copy number was estimated using the following formula:

$$\text{msDNA copy number} = 2^{(Ctg-Ctm)} - \text{plasmid copy number}.$$

This formula first estimates the total number of copies containing the msDNA sequence (which could be present on both plasmids and msDNA molecules). By subtracting the plasmid copy number (which also contains the msr/msd sequence), the msDNA copy number is isolated.

The copy numbers are relative estimates, assuming that the genomic DNA region has a copy number of 1. Due to potential differences in PCR amplification efficiencies between primers, these estimates are not absolute. Variations in amplification can affect the Ct values, so the plasmid and msDNA copy numbers should be interpreted as relative rather than precise measurements.

qRT-PCR was done on a Bio-Rad CFX96 instrument. Each reaction included: 10 μL Hieff® qPCR SYBR Green Master Mix, 1 μL each of forward and reverse primers (0.5 μM final concentration), 0.4 μL template DNA, nuclease-free water to a final volume of 20 μL. Thermal cycling conditions were: pre-denaturation at 95 °C for 4 min, denaturation at 95 °C for 10 s (repeated 40 cycles), annealing and extension at 60 °C for 30 s (during which fluorescence was measured). Melting curve analysis was done after 40 cycles to assess amplicon specificity. Amplification curves and melting curves were generated using CFX Maestro software 2.3 (Bio-Rad Laboratories, TX, USA). For each strain or control condition, three clones were used as biological replicates, with each sample run in triplicate wells for technical replication.

## Data availability

This study includes no data deposited in external repositories.

The source data of this paper are collected in the following database record: biostudies:S-SCDT-10_1038-S44319-024-00311-6.

## Peer review information

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

## Acknowledgements

This work was supported by the National Natural Science Foundation of China (31971339 and 32171422), the National Key Research and Development Program of China (2022YFF1000700), the Wuhan Natural Science Foundation Exploration Program (Chenguang Program) (2024040801020299), and the Fundamental Research Funds for the Central Universities (2662022SKYJ004). This work was also supported by the Science and Technology Research Project of Jiangxi Provincial Department of Education (Grant No: GJJ211722).

## Author contributions

**Yuyang Ni**: Data curation; Validation; Investigation; Methodology. **Yifei Wang**: Conceptualization; Data curation; Software; Investigation; Methodology. **Xinyu Shi**: Data curation; Validation; Investigation. **Fan Yu**: Investigation; Methodology. **Qingmin Ruan**: Data curation; Investigation. **Na Tian**: Validation; Investigation; Visualization. **Jin He**: Supervision. **Xun Wang**: Conceptualization; Supervision; Funding acquisition; Writing—original draft; Project administration; Writing—review and editing.

Source data underlying figure panels in this paper may have individual authorship assigned. Where available, figure panel/source data authorship is listed in the following database record: biostudies:S-SCDT-10_1038-S44319-024-00311-6.

## Disclosure and competing interests statement

The authors declare no competing interests.

# Expanded View Figures

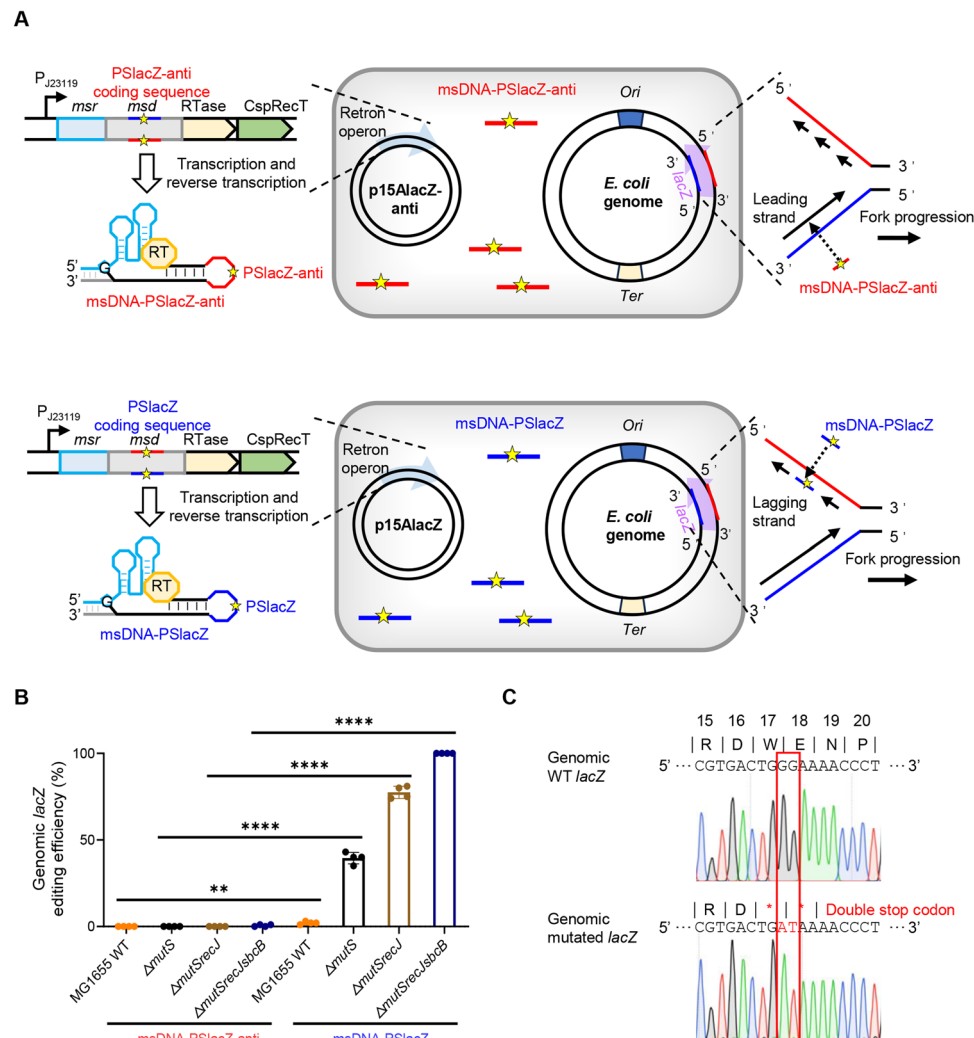

**Figure EV1. Schematic representation of the retron operon and msDNA-mediated genomic *lacZ* editing process.**

(A) Top panel: The retron operon is represented by double-stranded DNA labeled as the coding strand and template strand, respectively. A strong constitutive promoter, J23119 (black arrow), drives transcription of the operon, which encodes msDNA (*msr/msd*), reverse transcriptase (*ret*), and CspRecT (*recT*). The coding regions of *msr* and *msd* are depicted as blue and gray rectangles, respectively. A segment within the *msd* coding region (gray rectangle), which shares significant sequence similarity with the genomic *lacZ* gene, is denoted as the PSlacZ-anti coding sequence (PS stands for partial sequence). The blue line aligns with the *lacZ* coding strand, while the red line represents its reverse complement. The mutation sequence is indicated by a yellow star. Transcription and reverse transcription produce msDNA, a hybrid of msr RNA (light blue) and cDNA (black). The msDNA incorporates *lacZ* homologous sequences, designated PSlacZ-anti (red). *E. coli* genomic DNA and plasmid are shown as double-stranded structures. The genomic *ori* and *ter* are marked in blue and yellow, respectively. The *lacZ* gene and its transcription direction are indicated by a purple arrow. During *E. coli* genomic DNA replication, msDNA (red lines) integrates into the single-stranded DNA region at the replication fork through base pairing. Notably, msDNA-PSlacZ-anti has the same sequence as the leading strand. The detailed sequence of the plasmid is provided in Appendix Fig. S1. Bottom panel: The retron operon expresses msDNA-PSlacZ, which incorporates *lacZ* homologous sequences designated PSlacZ (blue). Importantly, PSlacZ is the reverse complement of PSlacZ-anti. PSlacZ shares the same sequence as the lagging strand. (B) Effect of strain and editing template on genomic *lacZ* gene editing efficiency at 24 h of incubation. We compared the editing efficiency in WT (no mutation control), Δ*mutS*, Δ*mutSrecJ*, and Δ*mutSrecJsbcB* strains expressing msDNA-PSlacZ-anti or msDNA-PSlacZ. The data are presented as a bar graph, with each bar representing the mean, and error bars indicating the standard deviation. Each data point represents an individual biological replicate ($n = 4$). Statistically significant differences were determined using an unpaired Student's *t*-test. The *p* value for the comparison of the editing efficiency between msDNA-PSlacZ-anti and msDNA-PSlacZ was 0.002714 in strain MG1655. In the Δ*mutS*, Δ*mutSrecJ*, and Δ*mutSrecJsbcB* strains, the *p* value for this comparison was <0.000001. **$P < 0.005$, ****$P < 0.0001$. (C) Sequencing results of the genomic *lacZ* sequence of the target region. Mutations change the codons for tryptophan and glutamate at positions 17 and 18 of the LacZ protein from TGG and GAA to stop codons (TGA and TAA). Red letters represent mutated bases. Source data are available online for this figure.

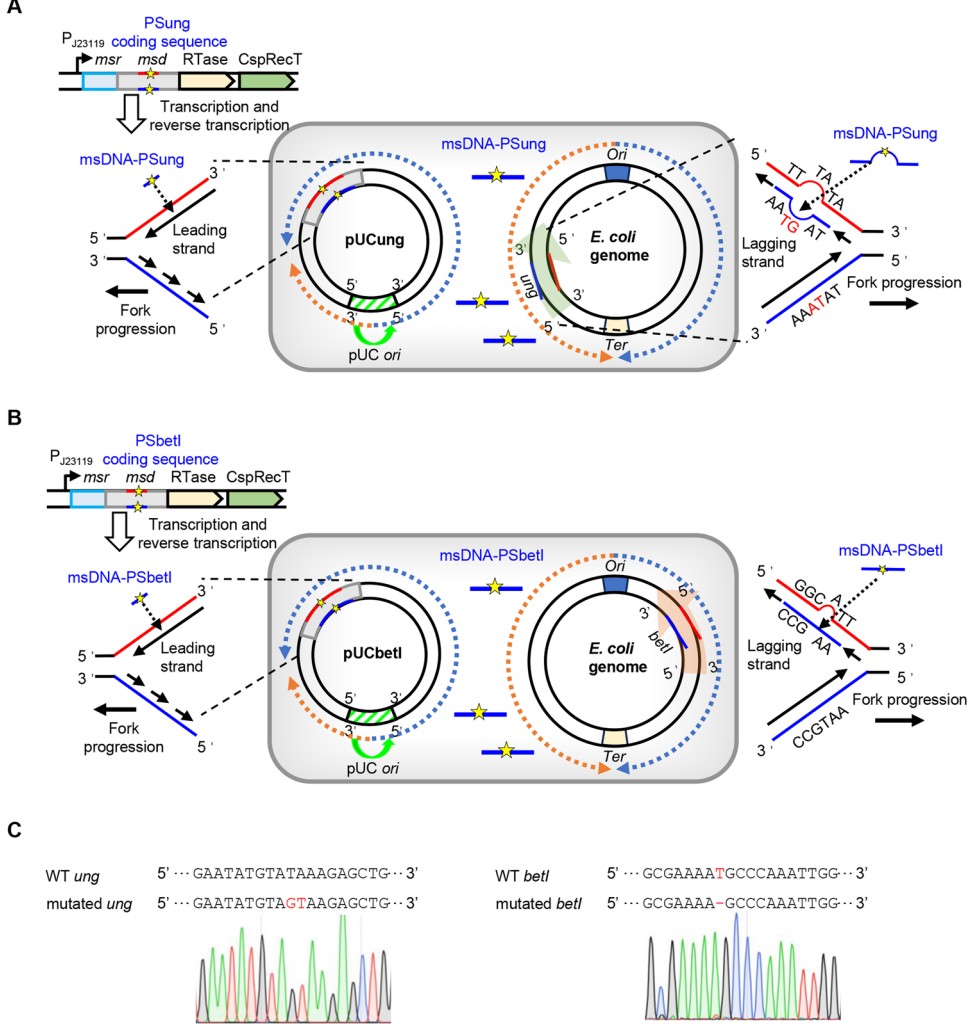

**Figure EV2. Genomic *ung* and *betI* gene editing in Δ*mutSrecJsbcB* strains harboring pUCung or pUCbetI plasmids, respectively.**

(A) A 92 bp homologous sequence of the *ung* gene (PS*ung* coding sequence) was inserted into the *msd* region. The msDNA-PSung, aligning with the genomic DNA lagging strand, mediates *ung* gene editing, resulting in double base mutations. Notably, the msDNA-PSung aligns with the pUCung plasmid's leading strand, preventing *msd* editing. The detailed sequence of the plasmid is provided in Appendix Fig. S13. (B) A 92 bp homologous sequence of the *betI* gene (PSbetI coding sequence) was inserted into the *msd* region. The msDNA-PSbetI, aligning with the genomic DNA lagging strand, mediates *betI* gene editing, resulting in one base deletion. Notably, the msDNA-PSbetI aligns with the pUCbetI plasmid's leading strand, preventing *msd* editing. The detailed sequence of the plasmid is provided in Appendix Fig. S14. (C) Sequencing results of the targeted region in the genomic *ung* and *betI* genes. Four colonies were randomly selected from each plate, and these colonies were pooled separately for each strain (one pool for Δ*mutSrecJsbcB*-pUCung and one pool for Δ*mutSrecJsbcB*-pUCbetI) to increase the number of cells analyzed for mutations. The regions surrounding the targeted sequences in the genomic *ung* and *betI* genes were separately amplified by PCR for each pool, followed by DNA sequencing. The DNA sequencing results reflect the average editing efficiency within each pool. If editing is highly efficient (close to 100%), we expect to see a dominant peak corresponding to the mutated sequence, with a very small or undetectable peak for the wild-type sequence. Conversely, lower editing efficiency will result in a mixture of peaks, representing both mutated and wild-type sequences. Mutated bases are indicated in red.

