## [Peer Review File · EMBO Reports]

Reducing competition between msd and genomic DNA improves retron editing efficiency

Yuyang Ni, Yifei Wang, Xinyu Shi, Fan Yu, Qingmin Ruan, Na Tian, Jin He and Xun Wang

Corresponding author(s): Xun Wang (wangxun@mail.hzau.edu.cn)

Review Timeline:

Submission Date:	6th Aug 24
Editorial Decision:	18th Sep 24
Revision Received:	7th Oct 24
Editorial Decision:	16th Oct 24
Revision Received:	20th Oct 24
Accepted:	25th Oct 24

Editor: Achim Breiling

Transaction Report:

Dear Dr. Wang,

Thank you for the submission of your manuscript to EMBO reports. I have now received the reports from the three referees that were asked to evaluate your study, which can be found at the end of this email.

As you will see, the referees have several comments, concerns, and suggestions, indicating that a major revision of the manuscript is necessary to allow publication of the study in EMBO reports. As the reports are below, and all the concerns need to be addressed, I will not detail them further here.

Given the constructive referee comments, I would like to invite you to revise your manuscript with the understanding that the concerns of the referees must be addressed in the revised manuscript and in a detailed point-by-point response. Acceptance of your manuscript will depend on a positive outcome of a second round of review. It is EMBO reports policy to allow a single round of revision only and acceptance of the manuscript will therefore depend on the completeness of your responses included in the next, final version of the manuscript.

- 1) a .docx formatted version of the final manuscript text (including legends for main figures, EV figures and tables), but without the figures included. Figure legends should be compiled at the end of the manuscript text.
- 2) individual production quality figure files as .eps, .tif, .jpg (one file per figure), of main figures and EV figures. Please upload these as separate, individual files upon re-submission.

- 4) a complete author checklist, which you can download from our author guidelines (<https://www.embopress.org/page/journal/14693178/authorguide>). Please insert page numbers in the checklist to indicate where the requested information can be found in the manuscript. The completed author checklist will also be part of the RPF.

- 5) that primary datasets produced in this study (e.g. RNA-seq, ChIP-seq, structural and array data) are deposited in an

appropriate public database. If no primary datasets have been deposited, please also state this in a dedicated section (e.g. 'No primary datasets have been generated and deposited'), see below.

The accession numbers and database should be listed in a formal "Data Availability" section (placed after Materials & Methods) that follows the model below. This is now mandatory (like the COI statement). Please note that the Data Availability Section is restricted to new primary data that are part of this study. This section is mandatory. As indicated above, if no primary datasets have been deposited, please state this in this section

Data availability

8) Regarding data quantification and statistics, please make sure that the number "n" for how many independent experiments were performed, their nature (biological versus technical replicates), the bars and error bars (e.g. SEM, SD) and the test used to calculate p-values is indicated in the respective figure legends (also for EV figures and all those in an Appendix). Please also check that all the p-values are explained in the legend, and that these fit to those shown in the figure. Please provide statistical testing where applicable. Please avoid the phrase 'independent experiment', but clearly state if these were biological or technical replicates. Please also indicate (e.g. with n.s.) if testing was performed, but the differences are not significant. In case n=2, please show the data as separate datapoints without error bars and statistics. See also: <http://www.embopress.org/page/journal/14693178/authorguide#statisticalanalysis>

9) Please also note our reference format:

10) We updated our journal's competing interests policy in January 2022 and request authors to consider both actual and perceived competing interests. Please review the policy <https://www.embopress.org/competing-interests> and update your competing interests if necessary. Please name this section 'Disclosure and Competing Interests Statement' and put it after the Acknowledgements section.

11) We now use CRediT to specify the contributions of each author in the journal submission system. CRediT replaces the author contribution section. Please use the free text box to provide more detailed descriptions and do NOT provide your final manuscript text file with an author contributions section. See also our guide to authors:

<https://www.embopress.org/page/journal/14693178/authorguide#authorshippinguidelines>

12) All Materials and Methods need to be described in the main text using our 'Structured Methods' format, which is required for all research articles. According to this format, the Materials and Methods section should include a Reagents and Tools Table (listing key reagents, experimental models, software, and relevant equipment and including their sources and relevant identifiers), uploaded as separate file, followed by a Methods and Protocols section in which we encourage the authors to describe their methods using a step-by-step protocol format with bullet points, to facilitate the adoption of the methodologies

across labs. More information on how to adhere to this format as well as downloadable templates (.doc) for the Reagents and Tools Table can be found in our author guidelines (section 'Structured Methods'):

Please order the manuscript sections like this, using these names:

Title page - Abstract - Keywords - Introduction - Results - Discussion - Methods - Data availability section - Acknowledgements (including funding information) - Disclosure and Competing Interests Statement - References - Figure legends - Expanded View Figure legends

I look forward to seeing a revised form of your manuscript when it is ready.

Yours sincerely,

Referee #1:

This manuscript reports a finding that is important to the community and would benefit from publication. I thank the authors for their efforts in producing this work.

I feel that the manuscript as written could be improved for additional clarity before publication, and have provided some suggestions below. I'd recommend the authors go beyond these suggestions to overall improve the clarity in which the work is presented prior to an additional round of review and/or publication.

My specific comments provided below:

In the abstract, the authors state "However, existing retron systems struggle with low efficiency." It's difficult to define what would be expected to be "low" or "high" and a continuous editing system like this could be run for few or many generations. Consider a more general statement like "existing retron systems deliver lower efficiency than is optimal for some applications" or "existing retron systems often deliver low editing efficiency".

Also in the abstract, when the authors state: "the msd gene, which encodes msDNA, 29 competes with the host's genomic DNA for limited msDNAs", this is potentially confusing, as the DNA element is not what is competing, but rather the RNA. Consider "the transcript of the msd gene which encodes msDNA" or similar.

In the abstract, the statement is made ". To overcome this hurdle, we engineered a novel retron system that 31 tailors msDNA specifically to the replicating leading strand of plasmids." While the observation contained here is important, I'm not sure it warrants being called a "novel retron system", because the retron being expressed is the same as in previous studies. Consider "we demonstrate important considerations for expressing retrons or similar elements to function at high efficiency" or similar.

In line 50 "producing a mutant-containing heterologous genome and a mutated genome" this may have to clarify "producing both mutated and wild-type progeny" or similar. Currently use of the word "heterologous" is contrasted to "mutant", which is incorrect. And this sentence implies a single mutant having both genomes, which is more confusing than focusing on the different genomes resulting from this editing event, wild-type and mutant.

In line 51 "Through the continuous triggering of mutations" - while persistently-expressed ssDNA may lead to continuous editing, electroporated ssDNA which is referenced in many of the prior references does not lead to continuous editing. The authors need

to clarify "persistent expression of ssDNA results in continuous production of mutations" or avoid this sentence altogether.

In line 62 "(previously ec86)." This should more clearly refer to Eco1, rather than BL21. Also, it's convention for these retron names to be capitalized. You can see the naming conventions as dictated by this paper:
<https://academic.oup.com/nar/article/47/21/11007/5584520>

In line 91 where two days is equated to 60 generations, it's best to state the generations and not the days, because it's the generations that matter most, and the method of growth can vastly change the number of generations per day

Line 107, "developing" implies a degree of novelty that I don't know is warranted here. Perhaps "designing"

The editing system described in the first/second paragraph of results would be expected to constitutively produce edits in LacZ resulting in a white phenotype. This will also occur while colonies are growing and developing. I'd like to see some mention of this technical detail at some point in the paper, and how white/blue and intermediate colonies are addressed and scored during experiments.

In line 136 "cannot be edited" should read "is edited at undetectable rate" or "is edited at low efficiency" or similar.

In the paragraph beginning in line 152, it should be clarified whether msDNA "expression" refers to the RNA transcript, or the successful production of msDNA.

For the experiment described in the paragraph starting in line 152, it should be stated that this method quantifies DNA copy relative to a genomic region, and the copy number of the genomic region is estimated to be 1 copy. This could be clarified either in the results or in the methods, but is important because these are not absolute measurements, but instead relative measurements.

LINE 162 "we found" should potentially read "we reasoned" as this is a hypothesis generated by reading the literature and considering mechanisms? This could be clarified in preceding sentences as well; potentially "studied" could be "considered" or similar.

Starting In line 183 the sentences including "based on these results we conclude"- the conclusion here may be overstated. I don't think this experiment proves the competition mechanism, but merely that additional copies reduce the efficiency. Perhaps "...the presence of additional homologous sequences on the plasmid reduce the efficiency of retron editing, in a manner consistent with these sequences competing with the genomic DNA target locus" or similar?

The sentence beginning in line 207 "these colonies..." may need to be clarified further. It's not clear what "further analysis" was performed, and what is being sequenced. As it currently reads, it's not clear that double-peaks or mixed bases would result only from editing. For instance, if the two plasmids were maintained in a strain and used as template for PCR, the PCR product would contain the sequences of both plasmids and result in mixed bases, even if no editing were to take place. This experiment needs further clarification, even if briefly.

The sentence ending in line 258 "...with varying colony sizes on" appears to be incomplete. Removing the word "on" could be sufficient, or more clarification needed.

The difference in growth rate explored in the paragraph beginning in line 257 is interesting- the authors should speculate as to the reason for this difference, even if it's merely speculation in the discussion section.

The sentence beginning in line 301 which states that editing was obtained within 2 hours or 6 generations needs to be clarified or altered. Because the system used is expressed continuously, and editing is measured by PCR of colonies seeded after this 2 hour incubation, editing is expected to occur while colonies are growing on the plate. In this experiment, it's not possible to discern whether the editing occurred in the brief 2-hour liquid culture, or in the time period wherein colonies were allowed to grow. This can be clarified by removing reference to specific numbers of generations and kept mostly as-is, the authors can make an estimate of the generations required to produce colonies during the experiment and add this to their numbers, or the authors can perform editing and measurement in liquid cultures, where the number of generations can be more accurately estimated.

Line 307 "increasing msd" needs to be clarified- probably increasing copy numbers of msd, or similar.

As stated previously, "novel retron system" in line 309 potentially should be restated, as this is an alteration and improvement to an existing retron system, in my opinion.

For the experiments described in "determination of plasmid and msDNA copy numbers" beginning in line 385, the Ct values of different amplicons of different sequence for genomic/msd sequences, produced using different primers are compared. This could result in differences in PCR amplification efficiency. This is unlikely to change the conclusions of the work, but is an

additional caveat when interpreting the absolute numbers which are presented. The authors should clarify this, at the very least by stating "an estimate of plasmid copy number" and "an estimate of msDNA copy number" rather than stating these as absolute values.

Referee #2:

The manuscript by Ni & Wang et al., describes that placing the msDNA in the leading strand of a plasmid reduce self-competition and highly increase editing rates using high-copy number plasmids, a key finding for retron-based recombineering technologies which could boost their use and efficiency. The authors first performed a set of confirmatory experiments that leads to the same conclusions as previous works in the field, but then, they cleverly show how retron msDNA self-targeting reduces editing efficiencies. The authors make the changes necessary in the plasmid/retron architecture to solve this issue leading to boosted editing rates using the retron machinery for short editing times.

Overall, this work will be of interest to the researchers in the field and to new readers interested in implemented the technology.

However, improving the clarity of the message and shortening the overall length could lead to a quick understanding of the manuscript and their key findings.

Thus, I would consider accepting the manuscript for publication in EMBO Reports if the authors consider addressing the points showed below:

-As a confirmatory result, Figure 1 could be moved to Supplementary Material and the corresponding writing section highly reduced and fused with current Section 2.

-Although visual, using clock times to describe the position of the origin of replication and the retron ncRNA in the different plasmids used in Figure 5 seems not reproducible when other plasmids with different origin of replications could be tested to find the best retron recombineering architecture. As the key finding in the manuscript, it could be crucial that the authors find a way to accurate describe these results.

-Editing new targets in the chromosome support the main finding, nevertheless, it is not necessary to make an additional result section to describe these results. These data could be described in a few sentences of the previous section to support the same finding and Figure 6 could be also move to Supplementary Material.

A few minor comments:

-Line 28: One of the limitations of retron recombineering is pointed out, but the msd region is stated as a gene, which is not accurate. Could the sentence be modified to describe the msd region properly?

-Lines 84-91: It could be worth citing Liu et al., 2023 reference in this paragraph, which also describe nearly 100% editing efficiency after optimization of strain, promoters, ribosomal binding sites and origin of replication (p15a) for retron recombineering.

Lines 92-96: To give further background of the efforts to increase retron editing rates in bacteria, it would be worth to also cite a recent work (Khan et al., 2024) which screens for new retrons finding new ones that outperform Eco1 efficiency.

Referee #3:

The rationale of Ni et al.'s work on retrons as genetic tools for genome editing is based on the observation that in plasmid constructs carrying the retron *msd* gene, which encodes retron msDNA, competition arises with the host's genomic DNA, significantly reducing editing efficiency. The authors thoroughly explored this issue through various experiments, including studying the effects of plasmid copy number on retron production and editing efficiency. Ultimately, they engineered a novel retron system that directs msDNA specifically to the replicating leading strand of plasmids, effectively preventing *msd* self-editing and eliminating competition for msDNA.

The manuscript is well-written and easy to follow overall, though certain expressions, such as "This ingenious..." in the abstract, feel somewhat out of place. The authors should be cautious with this type of language throughout the text.

General Comments:

I believe this manuscript will be of interest not only to specialists but also to general readers, as it clarifies many aspects that must be considered before using retron technology for editing prokaryotic genomes. However, their findings are not particularly surprising, as the bias toward the lagging strand has already been reported. It would be helpful if the authors could clarify how current published work using retron recombineering addresses the competition issue, placing their results more clearly in that context. Additionally, the manuscript does not address the application of retron technology in eukaryotes.

Other Comments:

1. In Figure 1 and the corresponding text, the type of plasmid used in the preliminary assays before testing different replicons is not mentioned.
2. The different constructs appear to have homologous sequences to *lacZ* around 92 nucleotides, but it is unclear which part of the retron *ec86* sequence is replaced.
3. Regarding the discussion on using retrons in eukaryotes, the authors should provide a clearer explanation of how self-targeting to the donor construct may affect the efficiency of the process.
4. It would be beneficial if the authors discussed recent findings from the Shipman lab, which show that some retrons with high efficiency for prokaryotic genome editing, due to large msDNA production, do not perform similarly in eukaryotes.

We sincerely thank all the referees for their encouraging and insightful feedback, which has greatly contributed to improving our manuscript. Below, we provide a detailed point-by-point response to the referees' comments:

Referee #1:

This manuscript reports a finding that is important to the community and would benefit from publication. I thank the authors for their efforts in producing this work.

I feel that the manuscript as written could be improved for additional clarity before publication, and have provided some suggestions below. I'd recommend the authors go beyond these suggestions to overall improve the clarity in which the work is presented prior to an additional round of review and/or publication.

My specific comments provided below:

In the abstract, the authors state "However, existing retron systems struggle with low efficiency." It's difficult to define what would be expected to be "low" or "high" and a continuous editing system like this could be run for few or many generations. Consider a more general statement like "existing retron systems deliver lower efficiency than is optimal for some applications" or "existing retron systems often deliver low editing efficiency".

Response: Thank you. We have revised the sentence to: "However, existing retron systems often exhibit suboptimal editing efficiency." This change is reflected in line 18 of the revised manuscript.

Also in the abstract, when the authors state: "the *msd* gene, which encodes msDNA, 29 competes with the host's genomic DNA for limited msDNAs", this is potentially confusing, as the DNA element is not what is competing, but rather the RNA. Consider "the transcript of the *msd* gene which encodes msDNA" or similar.

Response: Thank you for your feedback. We appreciate the opportunity to clarify this point. We would like to explain that the *msd* gene does compete with the host's genomic DNA for msDNA. This competition impacts the efficiency of genomic editing. To address potential confusion, we have revised the sentence to: "In *Escherichia coli*, we identified a critical bottleneck in the *msd* gene, which encodes the noncoding RNA template for msDNA synthesis and carries the homologous sequence of the target gene to be edited. This sequence homology causes the msDNA to bind to the *msd* gene, thereby reducing its efficiency in editing the target gene." This revision is reflected in lines 18-22.

In the abstract, the statement is made ". To overcome this hurdle, we engineered a novel retron system that 31 tailors msDNA specifically to the replicating leading strand of plasmids." While the observation contained here is important, I'm not sure it warrants being called a "novel retron system", because the retron being expressed is the same as in previous studies. Consider "we demonstrate important considerations for expressing retrons or similar elements to function at high efficiency" or similar.

Response: Thank you. We have removed the word “novel” and revised the sentence to read: “To address this issue, we engineered a retron system that tailors msDNA to the leading strand of the plasmid containing the *msd* gene.” Lines 22-24.

In line 50 “producing a mutant-containing heterologous genome and a mutated genome” this may have to clarify “producing both mutated and wild-type progeny” or similar. Currently use of the word “heterologous” is contrasted to “mutant”, which is incorrect. And this sentence implies a single mutant having both genomes, which is more confusing than focusing on the different genomes resulting from this editing event, wild-type and mutant.

Response: Thank you. We agree that the original phrasing could lead to confusion. To improve clarity, we have revised the sentence to: “In this model, ssDNA-binding proteins facilitate the alignment of the ssDNA donor with its complementary strand near the replication fork, creating a heteroduplex that includes a mutated strand. As this heteroduplex is replicated, both mutated and wild-type progeny are produced.” This revision is reflected in lines 38-41 of the revised manuscript.

In line 51 “Through the continuous triggering of mutations” - while persistently-expressed ssDNA may lead to continuous editing, electroporated ssDNA which is referenced in many of the prior references does not lead to continuous editing. The authors need to clarify “persistent expression of ssDNA results in continuous production of mutations” or avoid this sentence altogether.

Response: Thank you for your suggestion. We have removed the sentence “Through the continuous triggering of mutations...” in the revised manuscript.

In line 62 “(previously ec86).” This should more clearly refer to Eco1, rather than BL21. Also, it’s convention for these retron names to be capitalized. You can see the naming conventions as dictated by this paper: <https://academic.oup.com/nar/article/47/21/11007/5584520>

Response: Thank you for pointing this out. We have revised the sentence to: “One example is the Retron-Eco1 (Ec86) system discovered in *Escherichia coli* strain BL21.” Line 50 in the revised manuscript.

In line 91 where two days is equated to 60 generations, it’s best to state the generations and not the days, because it’s the generations that matter most, and the method of growth can vastly change the number of generations per day

Response: Thank you. We have revised the sentence to: “Other studies have demonstrated that knocking out *recJ* and *sbcB* in DH5 α strains, combined with the overexpression of Beta recombinase, can result in editing efficiencies nearing 100% in approximately 20 generations (about 16 hours) (Liu et al., 2023) or 60 generations (around two days) (Farzadfard et al, 2021).” Lines 78-81 in the revised manuscript.

In response to your suggestion, we have also revised similar formulations in the text to emphasize generations rather than days:

- “The gene editing efficiency of the first-generation system in the *E. coli* DH5 α strain is less than 10^{-4} after approximately 30 generations of induction (24 hours).” Lines 68-69.
- “Following approximately 20 generations (16 hours) of induction, the retron system’s editing efficiency improved 100-fold to 6×10^{-2} .” Lines 71-73.

Line 107, “developing” implies a degree of novelty that I don’t know is warranted here. Perhaps “designing”

Response: Thank you. In response to the second reviewer's suggestion to shorten the length of Result 1 and merge it with Result 2, we have combined these sections and updated the title to "Construction and evaluation of a retron editing system." Line 99 in the revised manuscript.

The editing system described in the first/second paragraph of results would be expected to constitutively produce edits in LacZ resulting in a white phenotype. This will also occur while colonies are growing and developing. I'd like to see some mention of this technical detail at some point in the paper, and how white/blue and intermediate colonies are addressed and scored during experiments.

Response: Thank you for your insightful comments. In our blue-white screening experiment, we count only fully edited bacteria from liquid culture, which form white colonies. If unedited bacteria grow during subsequent incubation on a solid medium, they may produce colonies with faint blue coloring. Therefore, any colony showing even a hint of blue is classified as unedited, while only completely white colonies are considered fully edited.

We have added these instructions to the Methods section, in lines 352-359. The details are as follows: We categorized the bacteria into three groups based on their editing status: Group 1 consists of fully edited bacteria, which are completely edited in the liquid medium, resulting in colonies that contain 100% non-functional *lacZ* and appear as white colonies. Group 2 includes partially edited bacteria that are unedited in the liquid medium but undergo editing while grown on solid medium, causing these colonies to exhibit a slight blue color due to the presence of unedited bacteria. Group 3 comprises unedited bacteria, which retain functional *lacZ* and thus form blue colonies due to X-gal hydrolysis. In our analysis, we recognized only the first group of bacteria as edited.

In line 136 "cannot be edited" should read "is edited at undetectable rate" or "is edited at low efficiency" or similar.

Response: Thank you. We have revised the sentence to: Our results demonstrate that msDNA efficiently edits the target gene only when the homologous sequence matches the lagging strand of genomic DNA replication, not the leading strand. Lines 117-119 in the revised manuscript.

In the paragraph beginning in line 152, it should be clarified whether msDNA "expression" refers to the RNA transcript, or the successful production of msDNA.

Response: Thank you. We have revised the sentence to clarify that "msDNA expression" refers to the successful production of msDNA. The revised sentence now reads: "These findings suggest that although msDNA production increases with plasmid copy number, it does not directly correlate with editing efficiency." Lines 134-136 in the revised manuscript.

For the experiment described in the paragraph starting in line 152, it should be stated that this method quantifies DNA copy relative to a genomic region, and the copy number of the genomic region is estimated to be 1 copy. This could be clarified either in the results or in the methods, but is important because these are not absolute measurements, but instead relative measurements.

Response: Thank you. We have clarified this in the methods section by adding the following statement: "The copy numbers are relative estimates, assuming that the genomic DNA region has a copy number of 1." This clarification has been added to lines 382-383 in the revised manuscript.

LINE 162 "we found" should potentially read "we reasoned" as this is a hypothesis generated by reading the literature and considering mechanisms? This could be clarified in preceding sentences as well; potentially "studied" could be "considered" or similar.

Response: Thank you. We have rewritten this paragraph as follows: “To understand the reduced editing efficiency observed in the *ΔmutSrecJsbCB*-pUClacZ strain, we considered the mechanism of the retron editing system. We noted that msDNA is synthesized using the retron operon as a template. This situation could create potential competition within the cell, as the synthesized msDNA could target two locations...” Lines 138-141 in the revised manuscript.

Starting In line 183 the sentences including “based on these results we conclude”- the conclusion here may be overstated. I don’t think this experiment proves the competition mechanism, but merely that additional copies reduce the efficiency. Perhaps “...the presence of additional homologous sequences on the plasmid reduce the efficiency of retron editing, in a manner consistent with these sequences competing with the genomic DNA target locus” or similar?

Response: Thank you for the suggestion. We have removed the word “conclude” and revised the sentence to: “This demonstrates that the 92 bp *lacZ* sequence on the plasmid was indeed edited by msDNA, and its presence reduces the efficiency of retron editing, likely due to competition between the plasmid-based sequences and the genomic DNA target locus for binding to the msDNA.” Lines 159-162 in the revised manuscript.

The sentence beginning in line 207 “these colonies...” may need to be clarified further. It’s not clear what “further analysis” was performed, and what is being sequenced. As it currently reads, it’s not clear that double-peaks or mixed bases would result only from editing. For instance, if the two plasmids were maintained in a strain and used as template for PCR, the PCR product would contain the sequences of both plasmids and result in mixed bases, even if no editing were to take place. This experiment needs further clarification, even if briefly.

Response: Thank you for your valuable feedback. We have revised the sentence for clarity: “We amplified the *msd* regions of the plasmids by PCR from the colonies of strain 1, using upstream primers located in the *sm^R* and *cm^R* gene regions, respectively, along with a downstream primer in the RTase gene region (see Appendix Table S1). This design ensured that only one plasmid served as the template for PCR amplification. Sequencing of the PCR product revealed a double peak at the 2 bp mutation site, indicating that the *msd* of the plasmid was successfully edited by msDNA (Fig. 3B).” Lines 183-188 in the revised manuscript.

The sentence ending in line 258 “...with varying colony sizes on” appears to be incomplete. Removing the word “on” could be sufficient, or more clarification needed.

Response: Thank you. We have removed the word “on”.

The difference in growth rate explored in the paragraph beginning in line 257 is interesting- the authors should speculate as to the reason for this difference, even if it’s merely speculation in the discussion section.

Response: Thank you for your insightful comment regarding the observed differences in growth rates of the *ΔmutSrecJsbCB*-pUClacZ2 strain. We propose that the observed slowdown in growth rate may be due to reduced plasmid stability, leading to a significant loss of plasmids. This loss could result in a sharp decrease in the number of antibiotic-resistant colonies, contributing to the observed decrease in growth rate. This discussion has been incorporated into lines 235-238 of the revised manuscript.

The sentence beginning in line 301 which states that editing was obtained within 2 hours or 6 generations needs to be clarified or altered. Because the system used is expressed continuously, and editing is measured by PCR of colonies seeded after this 2 hour incubation, editing is expected to occur while

colonies are growing on the plate. In this experiment, it's not possible to discern whether the editing occurred in the brief 2-hour liquid culture, or in the time period wherein colonies were allowed to grow. This can be clarified by removing reference to specific numbers of generations and kept mostly as-is, the authors can make an estimate of the generations required to produce colonies during the experiment and add this to their numbers, or the authors can perform editing and measurement in liquid cultures, where the number of generations can be more accurately estimated.

Response: Thank you. We have removed “6 generations” from the sentence. The revised sentence now reads: “Sequencing results revealed successful editing for both genes within two hours of liquid culture (Fig. EV2C).” Lines 252-253 in the revised manuscript.

Line 307 “increasing msd” needs to be clarified- probably increasing copy numbers of msd, or similar.

Response: Thank you. We have revised the text for clarity. It now reads: “increasing the copy number of msd” on line 258.

As stated previously, “novel retron system” in line 309 potentially should be restated, as this is an alteration and improvement to an existing retron system, in my opinion.

Response: Thank you. This sentence has been removed in the revised manuscript.

For the experiments described in “determination of plasmid and msDNA copy numbers” beginning in line 385, the Ct values of different amplicons of different sequence for genomic/msd sequences, produced using different primers are compared. This could result in differences in PCR amplification efficiency. This is unlikely to change the conclusions of the work, but is an additional caveat when interpreting the absolute numbers which are presented. The authors should clarify this, at the very least by stating “an estimate of plasmid copy number” and “an estimate of msDNA copy number” rather than stating these as absolute values.

Response: Thank you for your valuable feedback. We have added a clarifying sentence in the methods section: “Due to potential differences in PCR amplification efficiencies between primers, these estimates are not absolute. Variations in amplification can affect the Ct values, so the plasmid and msDNA copy numbers should be interpreted as relative rather than precise measurements.” This addition can be found in lines 383-386.

Furthermore, we have made several terminology changes throughout the manuscript to enhance clarity and accuracy: Changed “determine” or “calculate” to “estimate” (lines 361, 371, 374, 376, 377, 379, 382 and 384).

Referee #2:

The manuscript by Ni & Wang et al., describes that placing the msDNA in the leading strand of a plasmid reduce self-competition and highly increase editing rates using high-copy number plasmids, a key finding for retron-based recombineering technologies which could boost their use and efficiency. The authors first performed a set of confirmatory experiments that leads to the same conclusions as previous works in the field, but then, they cleverly show how retron msDNA self-targeting reduces editing efficiencies. The authors make the changes necessary in the plasmid/retron architecture to solve this issue leading to boosted editing rates using the retron machinery for short editing times.

Overall, this work will be of interest to the researchers in the field and to new readers interested in implemented the technology.

However, improving the clarity of the message and shortening the overall length could lead to a quick understanding of the manuscript and their key findings.

Thus, I would consider accepting the manuscript for publication in EMBO Reports if the authors consider addressing the points showed below:

-As a confirmatory result, Figure 1 could be moved to Supplementary Material and the corresponding writing section highly reduced and fused with current Section 2.

Response: Thank you for your suggestion. We have moved Figure 1 to Figure EV1. Additionally, we have combined Result 1 and Result 2 to streamline the writing.

-Although visual, using clock times to describe the position of the origin of replication and the retron ncRNA in the different plasmids used in Figure 5 seems not reproducible when other plasmids with different origin of replications could be tested to find the best retron recombineering architecture. As the key finding in the manuscript, it could be crucial that the authors find a way to accurate describe these results.

Response: Thank you for your insightful feedback. We have removed the clock time analogy and adopted a more precise description by specifying nucleotide positions. For instance, we clarified that the *msd* was relocated 991 bp clockwise from its original position to create the pUClacZ1 plasmid and an additional 827 bp clockwise for the pUClacZ2 plasmid (Lines 221 and 226).

In applying the conclusions to other plasmids with different origins of replication, it's important to note that, competition for msDNA due to plasmid replication is more pronounced in high-copy plasmids than in low or medium-copy plasmids. To reduce this competition in high copy number plasmids, we propose two specific strategies: 1) adjusting the relative position of *msd* on the plasmid, and 2) reversing the orientation of the plasmid replicon. For plasmids with a theta replication mechanism that replicates symmetrically (at the same rate in both directions), the first strategy is effective, while the second is not. However, the first approach may carry potential risks, such as altered msDNA production or reduced plasmid stability. In contrast, for plasmids that replicate at different rates in each direction, both strategies are effective. In these cases, we prefer the second method, as it reduces competition while minimizing the impact on plasmid stability and preserving the operon's function.

We have added the above content to the discussion section, specifically in lines 285-294 in the revised manuscript.

-Editing new targets in the chromosome support the main finding, nevertheless, it is not necessary to make an additional result section to describe these results. These data could be described in a few sentences of the previous section to support the same finding and Figure 6 could be also move to Supplementary Material.

Response: Thank you. We have merged the results of Figure 6 with those of Figure 5. Consequently, Figure 6 has been moved to the supplementary material as Figure EV2 in the revised manuscript (Lines 247-254 in the revised manuscript).

A few minor comments:

-Line 28: One of the limitations of retron recombineering is pointed out, but the *msd* region is stated as a gene, which is not accurate. Could the sentence be modified to describe the *msd* region properly?

Response: Thank you for your valuable feedback. We recognize the complexity of the retron system and would like to clarify the distinction between the *msd* gene and the *msd* region. The *msd* gene directs the synthesis of ncRNA, while the *msd* region refers to a sequence on the ncRNA that guides the synthesis of the cDNA portion of the msDNA (Lim and Maas, 1989; Mestre et al, 2020; Millman et al, 2020).

To enhance clarity, we have revised the abstract accordingly: In *Escherichia coli*, we identified a critical bottleneck in the *msd* gene, which encodes the noncoding RNA template for msDNA synthesis and carries the homologous sequence of the target gene to be edited. This sequence homology causes the msDNA to bind to the *msd* gene, thereby reducing its efficiency in editing the target gene. Lines 18-22 in the revised manuscript.

Lines 84-91: It could be worth citing Liu et al., 2023 reference in this paragraph, which also describe nearly 100% editing efficiency after optimization of strain, promoters, ribosomal binding sites and origin of replication (p15a) for retron recombineering.

Response: Done. Thank you.

Lines 92-96: To give further background of the efforts to increase retron editing rates in bacteria, it would be worth to also cite a recent work (Khan et al., 2024) which screens for new retrons finding new ones that outperform Eco1 efficiency.

Response: Thank you for the recommendation. We have added the citation for Khan et al. (2024) to lines 81-85 in the revised manuscript. The revised text now reads: “Additionally, a recent study screened for new retrons and identified several that outperform Eco1 in efficiency (Khan et al, 2024). This research highlights significant diversity in msDNA production and editing rates among retrons. These editors exceed those used in previous studies, achieving precise editing rates of up to 40% in human cells.”

Referee #3:

The rationale of Ni et al.’s work on retrons as genetic tools for genome editing is based on the observation that in plasmid constructs carrying the retron **msd** gene, which encodes retron msDNA, competition arises with the host’s genomic DNA, significantly reducing editing efficiency. The authors thoroughly explored this issue through various experiments, including studying the effects of plasmid copy number on retron production and editing efficiency. Ultimately, they engineered a novel retron system that directs msDNA specifically to the replicating leading strand of plasmids, effectively preventing **msd** self-editing and eliminating competition for msDNA.

The manuscript is well-written and easy to follow overall, though certain expressions, such as “This ingenious...” in the abstract, feel somewhat out of place. The authors should be cautious with this type of language throughout the text.

Response: Thank you for your valuable feedback and for recognizing the significance of our work. We appreciate your comments. We agree that certain phrases, such as “This ingenious...,” may not align with the scientific tone of the manuscript. In response to your suggestion, we have revised the abstract and other relevant sections to maintain a more neutral and objective tone, ensuring that the language remains appropriate throughout the manuscript.

General Comments:

I believe this manuscript will be of interest not only to specialists but also to general readers, as it clarifies many aspects that must be considered before using retron technology for editing prokaryotic genomes. However, their findings are not particularly surprising, as the bias toward the lagging strand has already been reported. It would be helpful if the authors could clarify how current published work using retron recombineering addresses the competition issue, placing their results more clearly in that context. Additionally, the manuscript does not address the application of retron technology in eukaryotes.

Response: Thank you for the opportunity to clarify the innovations in this manuscript. While previous studies have indeed reported a bias against the lagging strand, these studies primarily focused on the target gene being edited. Our research extends this by investigating the *msd* gene, which encodes the template for msDNA synthesis, providing new insights into the mechanics of retron editing. This adds a novel dimension to our understanding of how retron systems function.

We have incorporated this discussion into lines 259-265 in the revised manuscript as follows: “In conventional ssDNA-mediated gene editing, ssDNA is introduced into the cell via electroporation without requiring intracellular template synthesis. In contrast, retron editing synthesizes ssDNA within the cell using *msd* as a template. This introduces competition, which is absent in traditional ssDNA methods. Despite increasing interest in retron technology, prior studies have overlooked this competition issue. Our study is the first to identify and confirm *msd* editing by msDNA, and we propose a novel method to reduce this competition, significantly enhancing target gene editing efficiency.”

While our study focuses on *E. coli* as a model prokaryotic strain, we acknowledge the importance of applying retron technology to eukaryotic systems. In the revised manuscript, we have added a discussion on the potential use of retron systems in eukaryotes (lines 295-320 in the revised manuscript). Although further optimization is needed, retron technology has already been applied to eukaryotic cells for gene editing, laying the groundwork for future research in this area.

Other Comments:

1. In Figure 1 and the corresponding text, the type of plasmid used in the preliminary assays before testing different replicons is not mentioned.

Response: Thank you for your observation. We utilized the p15A plasmid in the preliminary assays. To enhance clarity, we have updated the graph to label the plasmids as p15AlacZ-anti and p15AlacZ (Figure EV1 in the revised manuscript). Additionally, we specified the plasmid type in the text, which now reads: “We constructed a retron editing system using a plasmid derived from the p15A origin of replication” (lines 100-101).

2. The different constructs appear to have homologous sequences to *lacZ* around 92 nucleotides, but it is unclear which part of the retron *ec86* sequence is replaced.

Response: Thank you for your attention to detail. Further information regarding the specific positions of the modifications within the *msd* sequence is provided in Appendix Figures S1 and S2, which clarify the parts of the retron ec86 sequence that were replaced in the different constructs.

To enhance clarity, we have revised the relevant sentence to: “The msDNA programmable region was modified with a 92 bp homologous sequences targeting either strand of *lacZ*, denoted as PSlacZ-anti and PSlacZ in Fig. EV1A and EV1B, respectively (where PS stands for partial sequence). Appendix Figs. S1 and S2 provide additional details on the specific locations of these modifications.” Lines 103-106 in the revised manuscript.

3. Regarding the discussion on using retrons in eukaryotes, the authors should provide a clearer explanation of how self-targeting to the donor construct may affect the efficiency of the process.

Response: Thank you. We have revised the discussion as follows: “To express msDNA in eukaryotic systems, retrons are typically integrated into shuttle plasmids, viral vectors, or directly into the host genome, resulting in multiple or single copies of *msd* within the cell. Studies in yeast have shown that ssDNA can participate in DNA replication as the lagging strand during plasmid replication, similar to the process observed in prokaryotes (Barbieri et al, 2017). This suggests that msDNA could potentially target originating plasmids or viral vectors and participate in their replication process. Such self-targeting may reduce the pool of msDNA available for effective HDR, thereby decreasing the overall efficiency of the editing process.” Lines 303-310 in the revised manuscript.

4. It would be beneficial if the authors discussed recent findings from the Shipman lab, which show that some retrons with high efficiency for prokaryotic genome editing, due to large msDNA production, do not perform similarly in eukaryotes.

Response: Thank you for your suggestion. We have included a discussion in lines 313-320 of the revised manuscript, which now reads: “Furthermore, certain retron editing systems that demonstrate high efficiency in prokaryotes do not yield similar results in eukaryotic cells (Khan et al, 2024; Lopez et al, 2022). We believe this difference arises from variations in editing mechanisms. In prokaryotes, msDNA is utilized during DNA replication for editing, whereas in eukaryotes, it serves as a template for repair during the HDR process following DSBs. Therefore, in addition to the competitive mechanisms revealed in this study, we speculate that another significant factor limiting the efficiency of retron editing in eukaryotes is the inefficiency of HDR.”

Dear Dr. Wang,

Thank you for the submission of your revised manuscript to our editorial offices. I have now received the report from the three referees that were asked to re-evaluate the study, you will find below. As you will see, the referees now fully support the publication of the study in EMBO reports.

Before I can proceed with formal acceptance, I have these editorial requests I ask you to address in a final revised manuscript:

- Please remove the word 'dramatically' from the title.
- Please provide the abstract written in present tense throughout.
- We plan to publish your manuscript as Report, as there are only 4 main figures and 2 EV figures. However, for a Scientific Report we require that results and discussion sections are combined in a single chapter called "Results & Discussion". Please do this for your manuscript. For more details please refer to our guide to authors:
<http://www.embopress.org/page/journal/14693178/authorguide#researcharticleguide>
- Please provide individual production quality figure files as .eps, .tif, .jpg (one file per figure), of the main and EV figures. Please upload these as separate, individual files upon re-submission, without their legends.
- Please remove the 'Materials availability' statement. Please note our guidelines:
<https://www.embopress.org/page/journal/14693178/authorguide#availabilityofpublishedmaterial>
- Please make sure that the number "n" for how many independent experiments were performed, their nature (biological versus technical replicates), the bars and error bars (e.g. SEM, SD) and the test used to calculate p-values is indicated in the respective figure legends. Please also check that all the p-values are explained in the legend, and that these fit to those shown in the figure. Please provide statistical testing where applicable. Please avoid the phrase 'independent experiment', but clearly state if these were biological or technical replicates. Please also indicate (e.g. with n.s.) if testing was performed, but the differences are not significant. In case n=2, please show the data as separate datapoints without error bars and statistics. See also:
<http://www.embopress.org/page/journal/14693178/authorguide#statisticalanalysis>

If n<5, please show single datapoints for diagrams. Moreover:

- Please define the error bars in the legends of figures 1C and EV1B.
- 1C does not show a scatter blot (as indicated in the legend), but a bar diagram.
- Please add statistics to the blots shown in 1B and EV1B.
- Please use our reference format (for more than 10 authors please use et al.):
<http://www.embopress.org/page/journal/14693178/authorguide#referencesformat>

- All Materials and Methods need to be described in the main text using our 'Structured Methods' format, which is required for all research articles. According to this format, the Methods section should include a Reagents and Tools Table (listing key reagents, experimental models, software, and relevant equipment and including their sources and relevant identifiers), uploaded as separate file, followed by a Methods section in which we encourage the authors to describe their methods using a step-by-step protocol format with bullet points, to facilitate the adoption of the methodologies across labs. More information on how to adhere to this format as well as downloadable templates (.doc) for the Reagents and Tools Table can be found in our author guidelines (section 'Structured Methods'):

<https://www.embopress.org/page/journal/14693178/authorguide#structuredmethods>

Thus, please remove the table from the methods section and upload it separately and providing callouts. Moreover, please add the primer information to this table and remove it from the Appendix.

- Please provide the Appendix file as PDF. Moreover, please use the nomenclature "Appendix Figure Sx" each figure's title legend throughout the file and for their callouts.

- Thank you for providing the requested source data (SD). Please upload this separated, with one folder per main figure (with all files for one figure in one folder and ZIPed) and one combined folder with the SD for EV or Appendix figures.

In addition, I would need from you uploaded separately:

- a short, two-sentence summary of the manuscript (not more than 35 words).
- two to four short (!) bullet points highlighting the key findings of your study (two lines each).
- a schematic summary figure as separate file that provides a sketch of the major findings (not a data image) in jpeg or tiff format

(with the exact width of 550 pixels and a height of not more than 400 pixels) that can be used as a visual synopsis on our website.

Best,

Referee #1:

Previously I helped submit recommendations for revision for this article. The authors have responded adequately to all my specific concerns, and I think the article is fit for publication in your journal.

Referee #2:

The authors have satisfactorily answered my comments and the manuscript was improved in the revision process. Then, I support the publication of this work in EMBO reports.

Referee #3:

The authors have satisfactorily addressed all the points raised in the first draft. I believe the manuscript has been significantly improved and is now suitable for publication.

All editorial and formatting issues were resolved by the authors.

Xun Wang
Huazhong Agricultural University
State Key Laboratory of Agricultural Microbiology
No. 1 Shizishan Street
Wuhan, Hubei 430070
China

Dear Dr. Wang,

I am very pleased to accept your manuscript for publication in the next available issue of EMBO reports. Thank you for your contribution to our journal.

Yours sincerely,
